# Programming gene expression in multicellular organisms for physiology modulation through engineered bacteria

Baizhen Gao[1] & Qing Sun [1 ✉]

A central goal of synthetic biology is to predictably and efficiently reprogram living systems to perform computations and carry out specific biological tasks. Although there have been many advances in the bio-computational design of living systems, these advances have mainly been applied to microorganisms or cell lines; programming animal physiology remains challenging for synthetic biology because of the system complexity. Here, we present a bacteria-animal symbiont system in which engineered bacteria recognize external signals and modulate animal gene expression, twitching phenotype, and fat metabolism through RNA interference toward *gfp*, *sbp-1*, and *unc-22* gene in *C. elegans*. By using genetic circuits in bacteria to control these RNA expressions, we are able to program the physiology of the model animal *Caenorhabditis elegans* with logic gates. We anticipate that engineered bacteria can be used more extensively to program animal physiology for agricultural, therapeutic, and basic science applications.

[1] Department of Chemical Engineering, Texas A&M University, College Station, TX, USA. ✉email: sunqing@tamu.edu

Recent advances in synthetic biology have resulted in highly programmable bacteria and mammalian cells that perform specific functions[1]. At the bacterial level, researchers have created logic functions and memory in living bacteria for bacteria and community oscillation, programmed bacteria to respond to signals, automated the circuit design process, and expanded the repertoire of logic gates so that they can recognize a variety of signals, including light and nutrients[1–3]. At the level of mammalian cells, researchers have engineered CAR-T cells and other mammalian cell types with switching logic circuits to program cell performance[4–8]. Despite rapid advances and promising results in bio-computational design, the programming of higher organisms remains largely unexplored because the engineering of an entire organism requires highly complex and sometimes impossible tasks, such as the delivery of gene editing cascades[9–11] or the engineering of whole embryos[12,13]. Nevertheless, such programming is worthwhile because it will yield a greater understanding than we have so far of how biological phenomena emerge from genetic and cellular events. The ability to change animal including insects and worm physiology would open up possibilities for agriculture, biomedical, and bioenvironmental applications.

Animals share a life-long relationship with microbes, which play significant roles in animal nutrition, immunity, behavior, and metabolism[14–17]. Engineering bacteria have demonstrated their potentials with pest control[18,19], plant growth promotion[20], human disease diagnosis[21–23], and therapeutics[24]. However, little has been done in terms of transferring the decision-making logics from bacteria to the animals to program animal physiology. Here, using bacteria as an engineering platform, we present an efficient strategy to program *Caenorhabditis elegans* GFP expression, twitching phenotype, and fat metabolism by building synthetic logic circuits in bacteria. We used the nematode *C. elegans* as the animal host as it has been extensively used as a model system to elucidate mechanisms of interaction between prokaryotes and their hosts[25,26]. The information from bacteria was transferred to *C. elegans* through an orthogonal gene transfer called RNA interference (RNAi), a widely used technique in which double-stranded RNA is exogenously introduced into an organism, causing knockdown of a target gene[27]. RNA interference is a regulatory mechanism conserved in eukaryotes that plays key roles in numerous biological processes, including RNA stability and processing, biotic and abiotic stress response, and the regulation of morphological and developmental events[28]. Horizontal transfer of mobile RNAs between different species has been observed between pathogens/parasites and host animals, pathogens/parasites and host plants, and plants and animals[28]. In *C. elegans*, RNAi is easy to implement because RNA can be delivered by feeding the worms bacteria that express double-stranded RNA complementary to a *C. elegans* gene of interest[29]. The inhibitory RNA is delivered from the bacterial cell to *C. elegans* when *C. elegans* grind on the bacteria by pharynx and absorb the bacterial contents[29]. The absorbed inhibitory RNA will then be distributed throughout the worm body leading to RNA interference[29,30].

In this work, we engineer bacteria to produce RNA, and by putting the RNA expression under the control of genetic circuits in bacteria, we program *C. elegans* green fluorescence protein (GFP) expression profiles, twitching phenotype, and worm fat storage with "AND" and "OR" logic gates. This study presents a platform for the programming of higher organisms by harnessing engineered prokaryotic species to exert effects on eukaryotic organisms (Fig. 1).

## Results

### Using RNA in ingested bacteria to silence GFP in *C. elegans* through RNAi.
We started with the genetically engineered strain *C. elegans* SD1084, which expresses nuclear-localized SUR-5:: GFP[31]. We intended to use the chemical inducer IPTG to silence GFP expression in this *C. elegans* strain through the worms' ingestion of engineered bacteria. To achieve that, we put *gfp* dsRNA synthesis under the control of a bidirectional lac promoter, which induces the synthesis of RNA in *E. coli* under IPTG induction. While many *E. coli* strains including HT115(*rnc-*), BL21(DE3), and OP50(*xu363*) have been utilized to produce RNA for RNAi in worm[32,33], within which HT115(*rnc-*) lacks RNase III to produce most dsRNA[32] and BL21 has the lowest RNA producing levels[32]. We started with BL21 because the relatively lower dsRNA production is correlated with less RNAi efficiency in *C. elegans* and thus can potentially provide a dynamic range of RNAi for a complete "ON" and OFF" state[32]. We used BL21(DE3) throughout this work. By feeding these induced plasmid carrying BL21(DE3) to *C. elegans*, we aimed to trigger *gfp* RNAi, which silences *gfp* in *C. elegans*[31] (Fig. 2a). We expected that the differences between the "ON" and "OFF" states of the RNAi in *C. elegans* would be large enough for us to measure. However, the first round of experiments with full-length *gfp* dsRNA led to *gfp* silencing under both the induced (i.e., with IPTG) and the uninduced (i.e., without IPTG) conditions (Supplementary Fig. 1). We hypothesized that the leaky lac promoter already produced enough dsRNA to silence the worm *gfp* without IPTG. Rather than solve the problem by tuning the strength of the lac promoter, which is a less desirable approach because tuning all promoters for genetic circuits in the future is highly laborious, we decided to work on the intermediates between the bacteria and the worm to achieve an appropriate dynamic range.

Systemic RNAi in *C. elegans* requires systemic RNA interference deficiency-1 (SID-1) protein for the transport of RNA[34,35]. SID-1 has been shown to have a lower binding affinity for shorter RNA sequences[34] leading to lower RNAi efficiency. We hypothesized that we could achieve the right dynamic range of RNAi by reducing the length of the RNA and thus transporting less RNA into the host with less RNAi efficiency. We constructed plasmids for RNA synthesis with lengths ranging from 100 bp to 400 bp in double-stranded as well as single-stranded format (Supplementary Fig. 2). After feeding the bacteria containing these plasmids to the worms, we found that single-stranded RNA sequences of 200 bp, 300 bp, and 400 bp achieved distinguishable "ON" and "OFF" status of GFP expression in *C. elegans* with and without IPTG (Fig. 2b). We picked the single-stranded 200 bp *gfp* RNA for subsequent experiments (Fig. 2c).

The efficiency of RNAi can be manipulated by changing the RNA concentration[27,34], which can be controlled by the concentration of inducers. Using the 200-bp design, we tested the dynamic range of the RNAi. We achieved a silencing of the GFP in *C. elegans* along a gradient, such that 81% of GFP in *C. elegans* was silenced when 15 µM of IPTG was supplied on the plate (Fig. 2d and Supplementary Fig. 3A). To further support our observation on the dose-dependent silencing effect, quantitative reverse transcription PCR (RT-qPCR) was used to measure the RNA synthesis levels at different IPTG concentrations. The bacterial RNA concentration is consistent with the worm RNAi efficiency (Supplementary Fig. 3B). What is worth mentioning here is that the *C. elegans* strain SD1084 we picked express nuclear-localized SUR-5::GFP[31] in all cells with a flag-tagged GES-1 (a carboxylesterase) protein co-overexpressed[31]. Since neuronal genes are more difficult to target with RNAi[36], we usually see ~20% residual fluorescence on our "OFF" stage, which should come from the neuronal GFP that were not RNAi silenced.

**Programming GFP expression with logic gates.** Logic gates, the building blocks of genetic circuits, can process complex signals and trigger downstream reactions[1]. Here we put *gfp* RNA

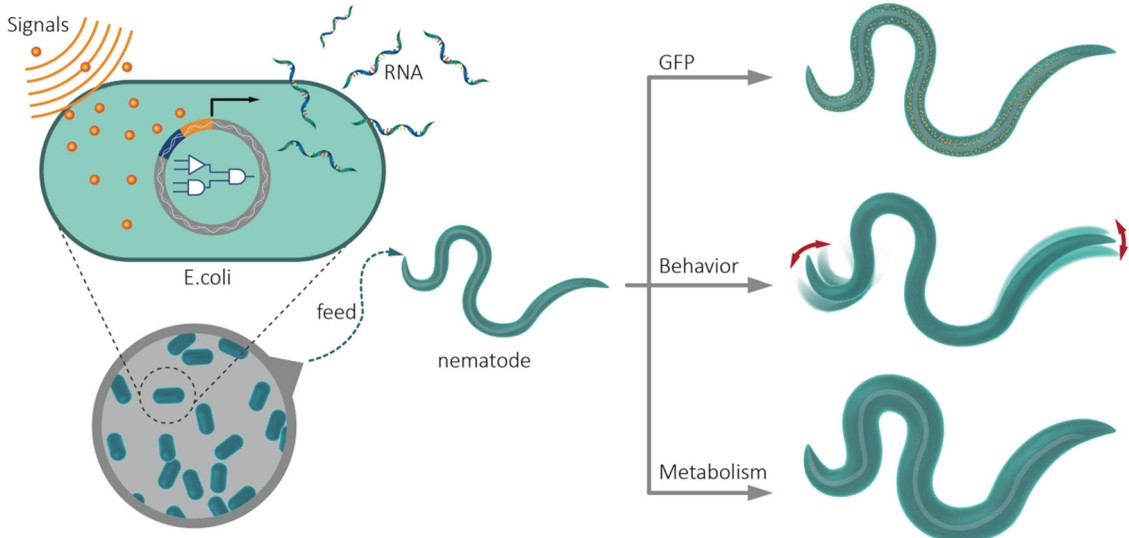

**Fig. 1 Engineering bacteria to program *C. elegans* GFP expression and physiology.** We have engineered bacteria to produce RNA, and by putting the RNA expression under the control of genetic circuits in bacteria, we programmed *C. elegans* green fluorescence protein (GFP) expression profiles, fat storage, and twitching phenotype with "AND" and "OR" logic gates.

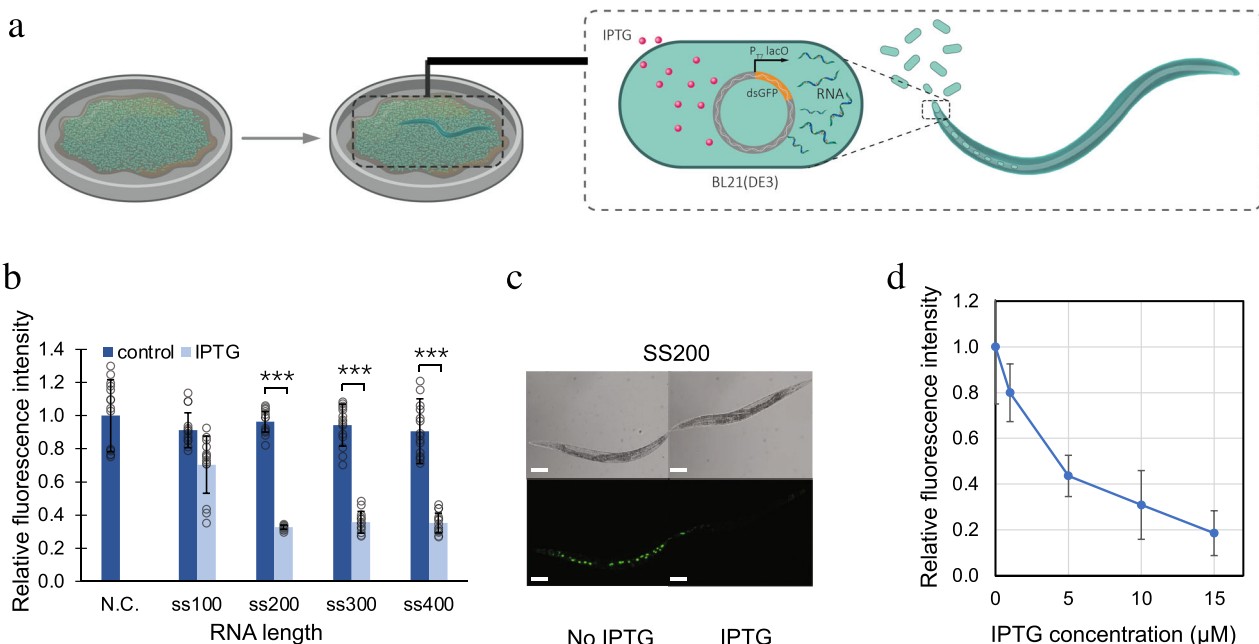

**Fig. 2 Using RNA in ingested bacteria to silence GFP in *C. elegans*. a** Schematic of engineered *E. coli* triggering RNAi in *C. elegans* through feeding. **b** Quantification of *C. elegans* GFP fluorescence intensity controlled by bacteria yielded single-stranded *gfp* RNA of various lengths with no IPTG and 1 mM IPTG. **c** Optical and fluorescent images of *C. elegans* SD1084 GFP expression programmed by bacterial single-stranded 200 bp *gfp* RNA (ss200) with no IPTG and 1 mM IPTG. Scale bar, 100 μm. The experiment was repeated three times independently for a total of 15 worms with similar results. **d** *C. elegans* SD1084 GFP intensity change caused by varying IPTG concentration-induced RNA synthesis. (***$P < 0.001$ using two-sided *t*-test. Error bars represent the mean ± standard deviation for $n = 15$ worms over three independent experiments). Source data are available in the Source Data file.

synthesis under the control of an "AND" or an "OR" gate and transferred the programming languages to *C. elegans* through RNAi.

An "AND" gate can be built by utilizing the split T7 polymerase (T7 RNAP)[37]. T7 RNAP is a single-subunit RNA polymerase that drives transcription by acting on its cognate promoter, the T7 promoter PT7. By splitting T7 RNAP between amino acids 179 and 180, a transcriptional AND gate is created whereby both fragments of the split protein are needed to drive transcription from PT7[37].

We used two orthogonal promoters, pBAD and pTet, to control T7 RNAP N-terminal and C-terminal expression, respectively. Both L-arabinose (Ara) and anhydrotetracycline (aTc), the respective inducers of these promoters, are needed to induce synthesis of *gfp* ssRNA coding strand fragment in the bacteria. The *gfp* RNA fragment will then lead to RNAi in the worm after the worm grinds the bacteria open and absorb the RNA fragment. With this system, worm GFP expression was silenced by >70% in the presence of both signals (Ara and aTc) whereas GFP was still expressed (>95%) with

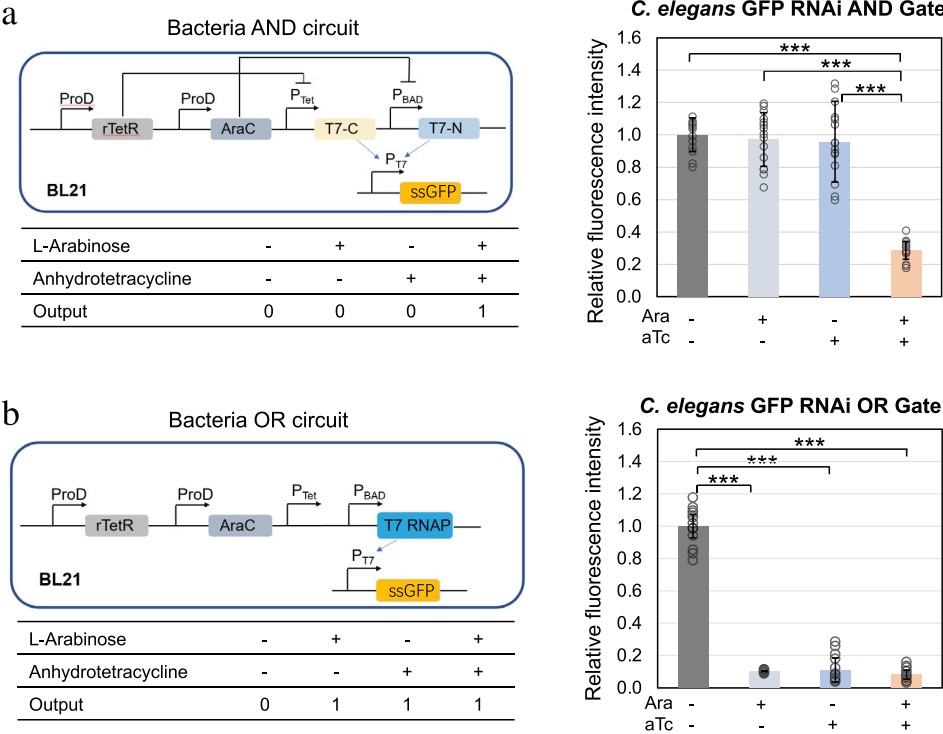

**Fig. 3 GFP expression in *C. elegans* programmed by AND and OR gates.** Single-stranded 200-bp *gfp* RNA synthesis in *E. coli* BL21, controlled by **a** AND gate or **b** OR gate, interferes with GFP expression in *C. elegans* that have ingested the bacteria. (***P < 0.001 using two-sided *t*-test with Bonferroni correction. Error bars represent the mean ± standard deviation for n = 15 worms over three independent experiments) Source data are available in the Source Data file.

either no signal or only one of the two signals (Fig. 3a and Supplementary Fig. 4A). The RNA levels synthesized in the bacteria were consistent with the AND gate profile that was observed in the worm experiment (Supplementary Fig. 4C).

For the OR gate, the pTet and pBAD were arranged in tandem to control the synthesis of full-length T7 polymerase, which transcribes single-stranded RNA against GFP in engineered bacteria under the control of promoter PT7. Activation of either pTet or pBAD was expected to lead to GFP silencing in *C. elegans*. We observed that the addition of one or both of the inducers led to >90% reduction in *C. elegans* GFP intensity, while the no-inducer control group only had a 24% reduction (Fig. 3b and Supplementary Fig. 4B). Similar to the AND gate, the RNA levels measured by RT-qPCR was consistent with the OR gate profile (Supplementary Fig. 4D).

**Programmed *C. elegans* physiology with logic gates.** After demonstrating that *C. elegans* GFP expression could be programmed with bacterial genetic circuits, we programmed essential gene silencing that affects *C. elegans* physiology. First, we programmed the twitching phenotype in *C. elegans* by feeding *unc-22* RNA producing bacteria to worm to interfere with the UNC-22 protein expression through RNAi in the worm. Since *unc-22* encodes a polypeptide involved in locomotion[35], we were able to program the twitching phenotype in *C. elegans*. Using similar approaches as those described above for GFP, we screened out double-stranded 400 bp and 200 bp *unc-22* RNA for the "AND" and "OR" gates of worm twitching phenotype (Fig. 4a). The twitching phenotype was characterized by observing the *C. elegans* body movement under a microscope immediately after the worms were anesthetized by levamisole[38]. The AND gate was constructed by using the same genetic circuit-carrying plasmid and replacing the gene of interest with *unc-22*. Only the worms grown on plates with both inducers showed the twitching

phenotype. In all, 100% of the worms exhibited twitching phenotype in the presence of both signals while 0% of the worms on the other three plates (no inducers plate, aTc only plate, and Ara only plate) showed twitching phenotype. Using the same OR gate but replacing the GFP in the OR genetic circuits with *unc-22*, 100% of the worms grown on plates with one or both inducers showed twitching phenotype (Fig. 4b and Supplementary Fig. 5).

Besides twitching phenotype, we programmed *C. elegans* metabolism through the *sbp-1* gene. Previous studies have shown that *sbp-1* facilitates fat storage, and silencing *sbp-1* using RNAi leads to reduced body fat storage in *C. elegans*[39]. We modulated *C. elegans* fat storage by using *sbp-1* RNAi (Fig. 4c); the modulation of fat storage was characterized by changes in Nile red staining. Because full-length sbp-1 RNA modulated *C. elegans* fat storage with a clear "ON" and "OFF" status after inducer concentration optimization (Supplementary Fig. 6C), RNA length optimization experiments were not performed as in the *unc-22* experiment. When the worms grew on plates with bacteria containing plasmids for the AND gate and *sbp-1* RNA synthesis, only the group that had both inducers showed a significant reduction in both worm body size and fat storage as indicated by a reduction in Nile red staining. For the engineered OR gate, the group without inducers had a body size and a level of fat storage similar to those of normal *C. elegans* (Fig. 4D and Supplementary Fig. 6). All other groups, which were exposed to either one or both of the inducers, showed an over 60% reduction in fat storage, demonstrating the successful OR gate programming of *C. elegans* fat storage.

## Discussion
Although the technology for genetically engineering living organisms has advanced rapidly, the complex design of genetic circuits and gene editing in animals remain challenging because of the complexity of these systems. In contrast, genetic circuit

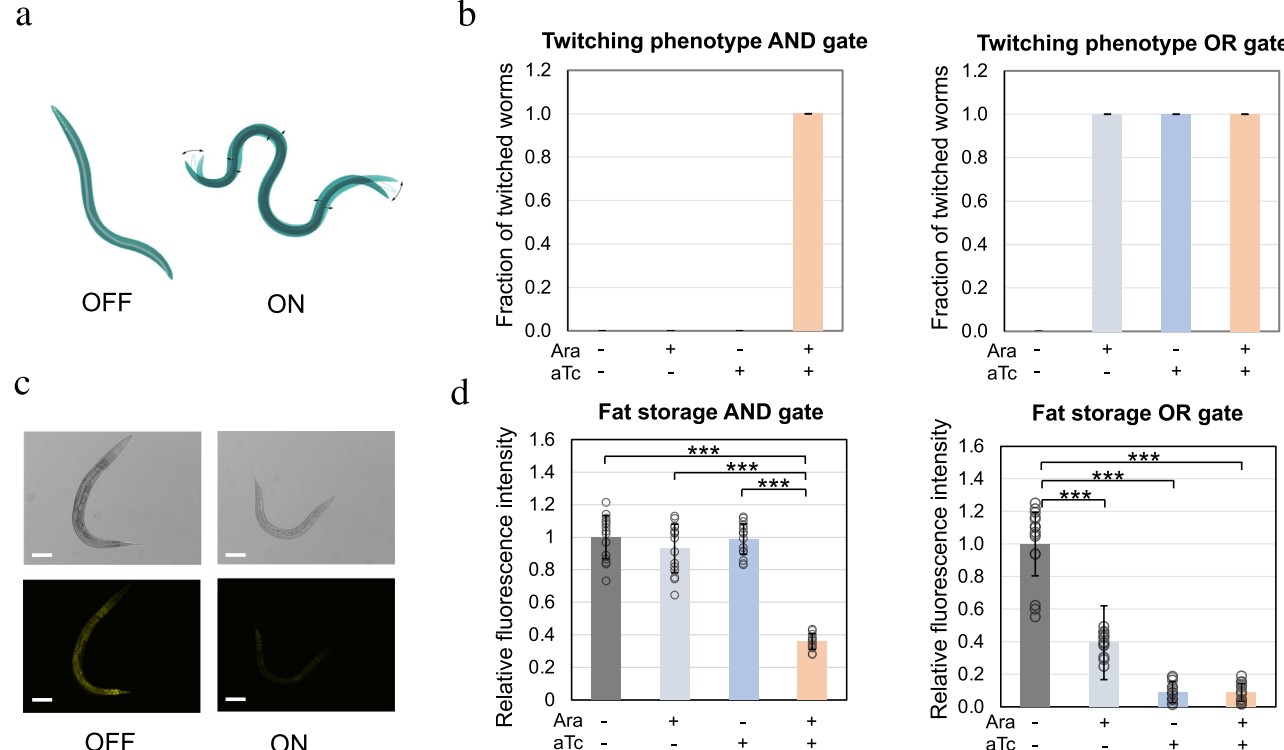

**Fig. 4 Worm twitching phenotype and fat storage programming through engineered bacteria. a** Schematic of twitching phenotype RNAi in *C. elegans*. **b** *C. elegans* twitching phenotype in response to AND and OR gate programming through bacteria expressing 200 bp and 400 bp double-stranded *unc-22* RNA. **c** Reduced fat storage due to *sbp-1* silencing, indicated by Nile Red staining. Scale bar, 100 μm. The experiment was repeated three times independently for a total of 15 worms with similar results. **d** AND and OR gate-controlled *C. elegans* metabolism through *sbp-1* RNA. (***$P < 0.001$ using two-sided *t*-test with Bonferroni correction. Error bars represent the mean ± standard deviation for $n = 200$ worms over three independent experiments for twitching phenotype, and $n = 15$ worms over three independent experiments for fat storage) Source data are available in the Source Data file.

design and editing have been highly successful in bacteria and mammalian cells. Bacteria affect the metabolism, immunity, and behavior of the animals they live in and on. We took advantage of the symbiosis between bacteria and animals to engineer prokaryotic cells as an indirect, alternative approach to programming animal physiology.

Using *E. coli* as the engineering platform and *C. elegans* as the host animal, we tuned *C. elegans* gene expression by changing the quantity of the RNA produced by the bacteria. The silencing of *C. elegans* gene expression was accomplished, through RNAi, with a reasonable dynamic range of programmable gene expression and physiology. We determined the effective dynamic range by screening out different lengths of RNA as mediators between bacterial and animal. An alternative way to tune the dynamic range is by changing the inducer concentration and incorporating T7 lysozyme to reduce basal RNA expression. This approach for basal level reduction as an alternative to shortening the RNA length is important because the logic gates can be tuned for enzyme expression rather than RNA synthesis. Furthermore, we applied genetic circuits to program *C. elegans* GFP expression, twitching phenotype, and fat storage through "AND" and "OR" logic gates. The fact that none of these worm phenotypes were affected by the inducers used in this study (L-arabinose and anhydrotetracycline) demonstrated that *C. elegans* underwent real programming by the engineered bacteria. The most striking advantage of this system is the ease with which system complexity can be extended by applying more complicated logic gates. Since bacteria have already been engineered with more complicated logic gates, memory, and sensing more signals, we should be able to program the animals with three or four input logic gates[3] or extend the capability of animals to recognize external signals[2,40,41] transmitted by these engineered bacteria.

RNAi is especially useful in the area of pests control and worm physiology interference as a gene silencing tool. However, for higher organisms in which RNAi would not be sufficient to alter physiology, bacterial metabolites could be exploited to serve as mediators between bacterial and animal (e.g., bile acid[42–44], butyrate[45–47], colonic acid[48], and nitrite[49–51]) to communicate between bacteria and higher organisms. Metabolites as intermediaries have broad applications for studying microbe-host symbiosis in situ; for example, they can be used as a drug delivery system. Indeed, bacterial metabolites operate as signaling molecules in the microbiome. Secondly, in this study we used synthetic inducers that were manually introduced into the host environment.

In summary, we have programmed *C. elegans* through the ingestion of engineered bacteria to exhibit changes in gene-expression and physiology. Our strategy could be extended to other bacteria or hosts, including fruit flies, zebrafish, plants, or mammals. The potential to program higher organisms using bacterial RNA or metabolites as intermediaries enabled us to study the bacteria-animal interaction.

## Methods

**Bacterial strains, worm strains, and growth conditions.** Bacterial strain *E. coli* BL21 (DE3) was used for testing of constructed plasmids encoding the RNA production without logic gate control. *E. coli* BL21 was transformed with both the constructed logic gate and RNA production plasmids. *E. coli* OP50 was used for

maintenance of *C. elegans*. Bacterial cultures were grown in LB media with appropriate antibiotics in a 37 °C shaking incubator overnight before use.

Wild-type *Caenorhabditis elegans* Bristol N2 and SD1084 (Caenorhabditis Genetics Center, Minnesota, USA) were cultured at room temperature on NGM (Nematode Growth Medium) agar plated with *E. coli* OP50 as described in WormBook[52]. Synchronized L1 worms were used in all RNAi treatment experiments. Synchronization was performed following the standard protocol[52]. Briefly, worms were allowed to grow on NGM plates with *E. coli* OP50 for 2–4 days then washed off the plates to collect a large number of gravid hermaphrodites, and eggs were isolated by bleach-NaOH lysis of the gravid worms. After 4X wash in M9 buffer to remove the residual bleach and NaOH, eggs were allowed to hatch in 10 mL of M9 buffer overnight and develop into starved L1, which were then used for the RNAi experiments.

**Plasmid construction for RNA synthesis and logic gates**. All plasmids were constructed using PCR and Gibson assembly and primers used are listed in Supplementary Table 1. Gene sequences used in this study are detailed in Supplementary Table 1. For double-stranded *gfp* and *unc-22* RNA synthesis, the RNA sequence of interest was inserted between two T7/LacO for the synthesis of both complementary strands on a pET24a vector. (Using this method, pET24a-dsgfp, pET24a-dsgfp-100bp, pET24a-dsgfp-400bp, pET24a-dsunc-22-200bp, and pET24a-dsunc-22-400bp were constructed.) Additional T7 lysozyme and LacI were constitutively expressed under the control of the proD promoter for double-stranded *sbp-1* RNAi to reduce background expression level for the construction of the plasmid pET24a-dssbp-1-proD-lysS-proD-lacI. We have added ProD promoter into the plasmid to overexpress promoter repressors, which improves the tightness of the pTet and pBAD promoters in the genetic circuit. We also used ProD to produce T7 lysozyme to reduce the background synthesis of RNA from lac promoter by binding with the T7 lysozyme to reduce the background transcription of RNA product[53,54].

Plasmids for single-stranded RNA synthesis were constructed by putting the RNA sequence of interest downstream of one T7/Lac on a pET24a vector. (Plasmids pET24a-ssgfp-100bp and pET24a-ssgfp-400bp were synthesized this way)

The AND gate plasmid (pTet-Ara-Split-T7-AND-Gate-proD) was modified from plasmid pTSlb-wt by constitutively expressing the TetR and araC repressors under proD promoters. The C-terminus of T7 RNA polymerase was regulated by pBAD, and the N-terminus of the T7 RNA polymerase was regulated by pTet. The OR gate (pTet-Ara-OR-Gate-proD) was constructed similarly by putting the full-length T7 RNA polymerase under the control of the two promoters arranged (pBAD and pTet) in tandem.

**Feeding *C. elegans* with the engineered bacteria for RNAi**. *E. coli* BL21 bacteria transformed with plasmids for RNA synthesis and logic gates were grown at 37 °C in LB with 50 µg/ml kanamycin and 25 µg/ml chloramphenicol, then seeded onto NGM plates supplemented with appropriate inducers as indicated in the supplementary information. After overnight growth of the seeded bacteria on NGM plates, synchronized L1 larvae were added to the plates and day-1 adult worms were used for phenotype assays after 2 days.

To program *C. elegans* GFP expression using IPTG, plasmids pET24a-T7-dsGFP, pET24a -dsgfp-100bp to pET24a-dsgfp-400bp were transformed into *E. coli* BL21(DE3) separately and fed to the worms. To induce the RNA synthesis in *E. coli*, 1 mM of IPTG was added into the NGM plates. To program GFP expression with an AND gate, *E. coli* BL21 was co- transformed with pTet-Ara-Split-T7-AND-Gate-proD and pET24a-ssgfp-200bp, and pTet-Ara-OR-Gate-proD and pET24a-ssgfp-200bp were co-transformed for OR gate programming. L-arabinose and anhydrotetracycline at 0.2 mg/mL and 0.1 µg/mL, respectively, were used as the external signal for both AND and OR gates.

For twitching phenotype AND gate programming, *E. coli* BL21 was co-transformed with pTet-Ara-Split-T7-AND-Gate-proD and pET24a-dsunc-22-400 bp. RNA synthesis was induced by 2 µg/mL L-arabinose and/or 0.1 µg/mL anhydrotetracycline. To program twitching phenotype using an OR gate, *E. coli* BL21 was co-transformed with pTet-Ara-OR-Gate-proD and pET24a-dsunc-22-200bp. RNA expression was induced by 0.2 mg/mL L-arabinose and/or 0.1 µg/mL anhydrotetracycline.

AND gate programming of fat storage in *C. elegans* was achieved by co-transforming *E. coli* BL21 with pTet-Ara-Split-T7-AND-Gate-proD and pET24a-dssbp-1-proD-lysS-proD-lacI, and expression was induced by 2 µg/mL L-arabinose and 0.1 µg/mL anhydrotetracycline. OR gate programming of fat storage was done by feeding the worms with *E. coli* BL21 co-transformed with pTet-Ara-OR-Gate-proD and pET24a-dssbp-1-proD-lysS-proD-lacI, and induced by 0.2 mg/mL L-arabinose and 0.1 µg/mL anhydrotetracycline.

For *gfp* and *sbp-1* RNAi experiments, five worms from each condition were analyzed in one experiment, and each experiment was repeated three times ($n = 15$ worms from each condition were analyzed in total)[31,39]. For the twitching phenotype, over 200 worms were analyzed from each condition. In each condition, we observed either twitching in all worms or no twitching at all. RNAi triggered 100% twitching phenotype was also observed in other study[32]. The number of

worms we used for calculation is in the same range as other paper when dealing with GFP quantification, fat storage, and twitching phenotype[31,39].

**C. elegans GFP fluorescence imaging**. Day-1 adult worms were rinsed off the NGM plates using PBS + 0.01% Triton X-100 (PBST) and washed twice with PBST to remove the bacteria. The worms were then transferred onto glass slides with 2% agarose containing 1 mM levamisole. Fluorescence in *C. elegans* was imaged on an Axiovert 200 M fluorescence microscope using the FTIC filter.

**Nile red staining**. *C. elegans* fat storage was characterized by Nile Red staining[55]. Synchronized L1 stage N2 worms were put onto the bacteria lawns, and allowed to grow for 2 days at room temperature. Young adult worms were washed off the plates by PBST and then fixed by 40% isopropanol (v/v) for 3 min. For lipid staining, a Nile Red stock solution of 0.5 mg/ml in acetone was first prepared. The working solution was then made by adding 6 µL of the stock solution into 1 mL of 40% isopropanol. After the worms were fixed and the supernatant removed, the working solution was added, and the worms were incubated in the working solution in the dark for 2 h. Following the 2-h incubation, the supernatant was again removed, 150 µL PBST was added, and the worms were incubated for another 30 min. The worms were then imaged using an Axiovert 200 M fluorescence microscope, and fluorescence intensity was quantified in ImageJ.

**Twitching phenotype recording**. After *C. elegans* were fed with the control and engineered *E. coli* for 2 days, the worms were washed off the plates using PBST. The worms were then washed once with PBST and resuspended with 20 µL 3 mM levamisole (in water)[38]. Resuspended worms were immediately transferred onto glass slides for recording under a microscope[38].

**RT-qPCR**. Synthesized single-stranded *gfp* RNA was also evaluated by RT-qPCR. Indicated bacteria was first allowed to grow overnight at 37 °C with 250 rpm shaking in LB medium with appropriate antibiotics. The overnight culture was then diluted 100 times the next day with supplementation of the appropriate inducers and antibiotics. The total RNA was extracted with RNeasy Mini Kit (Qiagen) when the O.D. reached 0.6–0.8. A total amount of 20 ng of the extracted RNA was then used for RT-qPCR by qScript 1-Step SYBR Green qRT-PCR Kit (Quantabio) on a LightCycler 96 (Roche). Each sample was measured with three biological replicates, and each biological replicate had three technical replicates on the 96-well plate. The primers used to quantify *gfp* RNA synthesis levels were gfp-F: 5′-CGGAGAAGAA CTTTTCACTGG-3′, and gfp-R: 5′-GGTAAGTTTTCCGTATGTTGC-3′. Synthesized gfp RNA levels were normalized to the expression of *cysG*[56] with the primers cysG-F: 5′-GATCGCGACTGTCTGATTG-3′, and cysG-R: 5′-CGGTGAACTGTG GAATAAACG-3′.

**Reporting summary**. Further information on research design is available in the Nature Research Reporting Summary linked to this article.

## Data availability
Data supporting the findings of this study are available within the paper and its Supplementary Information files. Source data are provided with this paper. The datasets generated and analyzed during the current study are available from the corresponding author upon request. Source data are provided with this paper.

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

## Acknowledgements

Lei Chen created Figs. 1, 2a, and 4a. This work is supported by Texas A&M University Experiment Station and Chemical Engineering Department Start-up funds.

## Author contributions

Q.S. conceived the study. B.G. and Q.S. designed the experiments. B.G. constructed plasmids and strains, and conducted all the worm-related experiments. Q.S. and B.G. wrote the manuscript.

## Competing interests

The authors declare no competing interests.
