## [Peer Review File · Nature Communications]

REVIEWER COMMENTS

Reviewer #1 (Remarks to the Author):

Review of: Programming animal physiology and behaviors through engineered bacteria.

Baizhen Gao and Qing Sun

This paper describes the engineering within *E. coli* of RNA expression designed to inhibit gene expression within *C. elegans*, and to control protein expression (constitutive GFP) or other genes regulating physiological functions such as twitch movement and fat storage.

The authors demonstrate that:

1. Exposure to an engineered *E. coli* that expresses iRNA to GFP results in reduction in GFP expression in *C. elegans* constitutively producing GFP. Inhibition of expression was under partial control of a lac promoter, inducible by exogenous IPTG. The promoter exhibited leakiness however, as GFP was inhibited even in the absence of IPTG (Figure 2B). The length of the RNA expressed had some impact on the level of inhibition but did not increase above 200 bp. IPTG concentration had a dose dependent effect on fluorescence intensity.
2. The authors take this an additional step forward from the engineering perspective and demonstrate that GFP expression can be controlled via AND and OR gates using two promoter inducer systems controlling a third gene for production of T7, an inducer of the PT7 promoter. Using a previously published split version of the T7 protein enable additional control by requiring both parts of the inducer to be made and bind to the promoter to turn on gene expression.
3. The authors further demonstrate control of endogenous *C. elegans* genes. The first is through protein production of the unc-22 protein, a polypeptide involved in locomotion. The second is through RNA inhibition of the spb-1 gene. Both are expressed using the AND and OR gating systems.

This paper demonstrates a technical achievement of utilization of synthetic biology and known promoter inducer systems to control gene expression in *E. coli* with an impact on *C. elegans* physiological functions. The data presented is clear, and the paper is well written. However, this work is not highly novel and the applicability to more complex eukaryotic systems is not adequately explored. Additionally there are some significant gaps in the work that would benefit from inclusion of additional explanation and data. These are described below:

Strain design, expression of effectors and release of these needs to be demonstrated. No data is included demonstrating the production and release of RNA from the cells. How is the inhibitory RNA being delivered from the bacterial cell? Is it secreted from the cell or released from cells that die or are engulfed by the host? The same questions apply to the protein secreting strain. How much unc-22 polypeptide is made, is it released from the cell into the media or stored intracellularly? More description of the choice of *E. coli* strain needs to be provided in the methods section. Is this a strain found in *C. elegans* or a laboratory strain of *E. coli*? Why was it chosen?

The authors claim they are feeding the *C. elegans* with this bacteria. It is not clear that the bacteria is being ingested by *C. elegans* from the media, or that the delivery of its products is occurring through this mechanism. The authors need to demonstrate how the effectors (RNA or protein) are being delivered to *C. elegans*. Is this occurring through the ingestion of the bacterial cells and activation from within the GI tract or are the molecules produced by the engineered cells being ingested or absorbed from the media by *C. elegans*? These are important questions to help better understand the relevance if this work for other orally delivered engineered microbes. Other developers of synthetically engineered microbes have surpassed the accomplishment of this work by demonstrating in vivo expression of engineered genes by the bacterial cells in mammalian systems including humans. These additional studies should be referenced. The AND/OR system in itself is not novel and it would have been more relevant if the chosen promoter inducer systems related to effectors that could only be produced, or conditions that could only turn on inside the worm. More discussion about how such AND/OR systems could be beneficial, rather than just simply turning on the gene in the host when it is needed, such as with a temperature or anaerobic promoter should be added to the discussion. Based

on these comments I would suggest the manuscript needs significant revision before resubmitting to Nature Communications for Review.

Some additional minor points on the manuscript are below:

- Please provide more detail in the abstract about which genes are being modulated in *C. elegans*
- Line 32, a word is missing in this sentence. "...because it will yield a greater understanding than we have so far of how biological phenomena..."
- Figure 2B,C: what concentration of IPTG was used for this study? Why is there no impact of IPTG in 2B but loss of GFP in 2C?
- The relevance of the size of the RNA fragment should be discussed
- Figure 3 and supplemental figures: In Figure 3B the output table for the OR circuit should this show a "1" when either L-Arabinose or Anhydrotetracycline are present? Check also the supplemental figures for similar error.
- Figure 4. The authors need to describe how many worms were counted in these studies. How many are enough to conclude 100% twitching?

Reviewer #2 (Remarks to the Author):

Gao and Sun describe genetic circuits that can induce RNAi in *C. elegans* by feeding bacteria. They engineer "AND" and "OR" logic gates into bacteria that synthesize single or double-stranded RNA, which are fed to worms to trigger RNAi. The bacterial genetic circuits utilize split or tandem T7 polymerases that conditionally activate T7 transcription in response to single or dual inducers (IPTG, Arabinose, and Tetracycline) to generate the AND and OR gates, respectively. The authors use this system to downregulate a GFP reporter, *unc-22*, and *sbp-1*. The authors claim that this represents an original example of programming *C. elegans* physiology, behavior, and metabolism through ingested bacteria. They also assert that their system explores the widely unexplored bio-computational design of higher organism, leading to metabolic and behavioral programming for agricultural, therapeutic, and basic science applications.

Major concerns

Genetic circuits are promising synthetic biology tools that have great potential in resolving the complexity of engineering and programming biological systems. However, their prior implementations have mostly been limited to bacteria and mammalian cell cultures. In principle, Gao and Sun present an elegant and novel approach to program higher multicellular organisms by exploiting their innate silencing mechanism and their relationship to bacterial symbionts. Programming a simple multicellular organism such as *C. elegans* by feeding bacteria would potentially be of great interest in the synthetic biology field. However, the examples that the authors develop are, unfortunately, trivial. If one accepts two well-established premises: (1) RNAi by feeding can induce phenotypes (e.g., Timmons and Fire, 1998 Nature and Fraser et al., 2000, Nature) and (2) bacteria can be engineered to express RNA in response to one or more inducers, then the authors merely recapitulate RNAi phenotypes that have been widely observed in hundreds of laboratories in response to RNAi by feeding. If animals are fed RNA targeting *gfp*, *unc-22*, or *sbp-1* then fluorescence is reduced, animals twitch in response to levamisole, or animals deposit more fat, respectively. So, in my view, the manuscript does not describe any new results or represent a true innovation.

For the manuscript to represent a real advance, I would expect the authors to engineer a new functionality into the organism and then control this functionality, e.g., with bacteria that express RNA in response to one or more inducers.

In my view, the authors also greatly overstate the importance of the advance. I do not believe that inducing simple phenotypes constitute "programming the physiology and behavior" of *C. elegans*. I am also not sure how the authors' system "can be developed for applications to pest control, disease diagnostics, and the delivery of therapeutics"?

General comments

From a more general perspective, I am not sure that RNAi by feeding is a particularly good choice for inducing phenotypes in response to external stimuli. The response times are slow and are poorly multiplexed with only partial effects when two genes are silenced and only rare effects when three genes are targeted. Importantly, genes expressed in neurons cannot be easily targeted (although this can be solved by engineering, see e.g., Firnhaber & Hammarlund 2013 PLoS Genetics) which would be the main tissue to for engineering "behavior". In the examples demonstrated by the authors, the output comes from disrupting genes during development. For most interesting behaviors, it would be desirable to rapidly induce, e.g., protein dimerization or protein degradation in response to light or chemical cues to change the behavior of a fully developed animal. These tools are widely available in other organisms and have also been used successfully in *C. elegans* (see e.g. Fielmich et al. 2018 Elife).

The authors argue that the use of short RNAs are necessary to achieve an appropriate dynamic range and that they "tune" the responses. But these observations almost certainly have to do with their initial choice of a leaky lac promoter. All the responses that the authors describe are "all or none" and do not involve any "tuning" of responses. In the absence of RNA expression from bacteria then animals will remain fluorescent, will not twitch, or deposit extra lipids. Once a threshold of RNA is produced, then the secondary small RNA amplification system (22G-RNAs, primarily) will kick in and silence the transgene or endogenous genes.

Despite my reservations about the current version of the manuscript, I do hope that the authors will persevere and further develop their system to truly engineer worms to control behaviors (very few would consider muscle spasms in response to a drug a "true" behavior). The presented results represent a good set of validation experiments that confirm their ability to reproduce previously published phenotypes. And the results would set the stage for truly engineering the behavior or physiology of a simple multicellular organism.

Specific comments and suggestions

The authors used a curious strain for the GFP experiments. SD1084 has a *sur-5::GFP* plasmid that expresses GFP in all cells but also has a flag-tagged *GES-1* (a carboxylesterase) protein overexpressed. The dynamic range of the GFP silencing is likely reduced because neuronal expression will not be silenced (which is likely why there is a 20% residual fluorescence). The strain background would not be expected to interfere with the responses but the authors may want to comment on this in the manuscript.

Lines 82-86 and Figure 2. It is curious, and unexpected, that single-strand RNA leads to silencing. Is the single-stranded DNA in fact two inverted repeats of GFP so a dumb-bell formed dsRNA is formed (see Timmons et al. 2001 for a configuration that works in this way)? If the fragment is only one strand of GFP then the figure labeling should be changed from dsGFP to ssGFP.

Also, the choice of RNAi bacteria is unusual. Almost all RNAi by feeding experiments are done using the HT115 strain that lacks a double-strand RNase activity (see same paper by Timmons et al.). The authors are likely to get a better dynamic range and more robust effects if using they switch to the canonical strain, unless HT115 cannot be engineered?

Please rephrase the sentence spanning lines 111 - 113. This sentence is confusing and inaccurate. The bacteria do not transcribe GFP (they make a fragment of *gfp*, and it's unclear if it is single stranded or double-stranded and if it is in the coding vs reverse direction). Also, RNAi is a process, so "RNAi expression" is not triggered in the worms.

Line 131 nomenclature: *unc-22* should be UNC-22 (when it is a protein). If you were referring to the gene, then it should be *unc-22* (in italics). This nomenclature should be corrected through-out the manuscript.

Statistics: It would be helpful if the authors indicate the number of experiments performed for each statistical comparison and whether graphs show standard deviations or standard error of means. Also, the statistical test (Student's T-test) used in figure 3 and 4 does not appear to have been corrected for multiple comparisons. The differences are robust and should easily hold up.

Reviewer #3 (Remarks to the Author):

In this work, the authors designed and engineered logic circuits in the model bacterium *Escherichia coli* to study programmable outputs in the physiology and behaviour of its host, *Caenorhabditis elegans*. The scope of this study is interesting and potentially a good foundational proof-of-concept study for using engineered bacteria to predict and program animal physiology for specific applications. However, the overall data are insufficient and do not fully support the conclusions of this study. The following comments should be addressed before the manuscript can be re-considered for publication.

Major Comments

1. The introduction section is inadequate in highlighting the background and importance of RNA in host-microbe interaction. Please highlight the importance more clearly to justify the direction and scope of this study.

2. The experimental rationale for RNA circuit design in *E. coli* is unclear and the following points should be addressed.

- Additional information on initial dsGFP RNA tested in figure S1 should be provided as this was not described sufficiently.

- In figure 1 and S2; it was hypothesized that the initial circuit did not produce the desired results as the lac promoter was leaky. This aspect was not addressed and subsequently, the same promoter was employed in downstream circuitry design as shown in figure 1 and S2. Please address the possibility, where promoter leakiness is the reason for the result obtained in Figure S1, and provide relevant data that either support or reject such a possibility.

- The authors' choice of reducing RNA length seems contradictory to the function of SID-1. Please explain clearly the link between SID-1's affinity for longer RNA and the choice for shorter RNA lengths.

3. In figure 3B; The OR gate output is inaccurate. The logical expression of OR gate in this context should be:

L-Ara aTC Output

- - 0

- + 1

+ - 1

+ + 1

where 0 = no GFP silencing and 1 = GFP silencing. This is also observed in Figure S4A and S5A. Please correct the OR gate outputs in these figures accordingly.

4. Please explain and justify the use of promoter ProD in the circuit design.

5. Optimization data were used to establish the combination of inducer concentrations to demonstrate OR and AND gate is not provided. In the same context, the arabinose concentrations used for AND gate and OR gate is significantly different. Please provide the necessary data and justify the inducer

concentrations used in this assay.

6. In figure 4B; the final unc-22 RNA used for figure 4B is not specified along with screening data for unc-22 RNAi candidates. Please provide necessary data to support the results in figure 4B with a detailed explanation for screening double-stranded RNA instead of single-stranded RNA for unc-22. This approach is contrary to earlier results shown in figure 1 and 2 where single-stranded RNA was used to establish circuit design.
7. In figure 4B; twitching behaviour observed shows high variability without quantitative and secondary qualitative measures. Please ensure the reliability and robustness of the data provided for the twitching behaviour observed along with providing error bars for the figure.
8. Screening and validation for sbp-1 RNAi candidates are lacking similar to unc-22. Please provide the required data.
9. The naming of parts in circuit design is inconsistent in the text leading to difficulties in understanding the circuit. For example, Page 6 lines 10 and 15 mentioned promoters control GFP transcription in engineered bacteria, this should rather be the transcription of single-stranded RNA against GFP in engineered bacteria. Please keep the naming of parts consistent in the text to avoid confusion.
10. In figure S4A & S4B; there is an incorrect citation of this figure in the text where figure S4A shows OR gate results while figure S4B shows AND gate results. Please revise accordingly.

Reviewer #1 (Remarks to the Author):

Review of: Programming animal physiology and behaviors through engineered bacteria.
Baizhen Gao and Qing Sun

This paper describes the engineering within *E. coli* of RNA expression designed to inhibit gene expression within *C. elegans*, and to control protein expression (constitutive GFP) or other genes regulating physiological functions such as twitch movement and fat storage.

The authors demonstrate that:

1. Exposure to an engineered *E. coli* that expresses RNA to GFP results in reduction in GFP expression in *C. elegans* constitutively producing GFP. Inhibition of expression was under partial control of a lac promoter, inducible by exogenous IPTG. The promoter exhibited leakiness however, as GFP was inhibited even in the absence of IPTG (Figure 2B). The length of the RNA expressed had some impact on the level of inhibition but did not increase above 200 bp. IPTG concentration had a dose dependent effect on fluorescence intensity.
2. The authors take this an additional step forward from the engineering perspective and demonstrate that GFP expression can be controlled via AND and OR gates using two promoter inducer systems controlling a third gene for production of T7, an inducer of the PT7 promoter. Using a previously published split version of the T7 protein enable additional control by requiring both parts of the inducer to be made and bind to the promoter to turn on gene expression.
3. The authors further demonstrate control of endogenous *C. elegans* genes. The first is through protein production of the *unc-22* protein, a polypeptide involved in locomotion. The second is through RNA inhibition of the *spb-1* gene. Both are expressed using the AND and OR gating systems.

This paper demonstrates a technical achievement of utilization of synthetic biology and known promoter inducer systems to control gene expression in *E. coli* with an impact on *C. elegans* physiological functions. The data presented is clear, and the paper is well written. However, this work is not highly novel and the applicability to more complex eukaryotic systems is not adequately explored. Additionally there are some significant gaps in the work that would benefit from inclusion of additional explanation and data. These are described below:

Dear reviewer, thank you for your comments concerning our manuscript entitled "Programming animal physiology and behaviors through engineered bacteria". These comments are very valuable and insightful. We studied your comments thoroughly and have made corrections/responses as follows.

Strain design, expression of effectors and release of these needs to be demonstrated.

No data is included demonstrating the production and release of RNA from the cells. How is the inhibitory RNA being delivered from the bacterial cell? Is it secreted from the cell or released from cells that die or are engulfed by the host?

- Thank you for your question about RNA production and delivery from the bacteria to *C. elegans*. The inhibitory RNA is delivered from the bacterial cell to *C. elegans* when *C. elegans* grind on the bacteria by pharynx and absorb the bacterial contents¹. The absorbed inhibitory RNA will then be distributed throughout the body leading to RNA interference (RNAi)^{1,2}. Compared with injecting RNA directly, bacteria can sense the environment, make decisions, and produce the RNA for decision transfer purposes. The efficiency of the RNAi increased with the amount of RNA produced and fed number of the bacteria (RNA)³. The amount of synthesized RNA can be quantified by quantitative reverse transcription PCR (RT-qPCR). We also added the RT-qPCR results indicating the bacterial RNA synthesis in the manuscript in Fig. S3B, Fig. S4C, and Fig. S4D.

To address this question, we have added the following information in line 49-52: “The inhibitory RNA is delivered from the bacterial cell to *C. elegans* when *C. elegans* grind on the bacteria by pharynx and absorb the bacterial contents¹. The absorbed inhibitory RNA will then be distributed throughout the worm body leading to RNA interference^{1,2}.”

Line 107-109: “To further support our observation on the dose-dependent silencing effect, quantitative reverse transcription PCR (RT-qPCR) was used to measure the RNA synthesis levels at different IPTG concentrations. The bacterial RNA concentration is consistent with the worm RNAi efficiency (Fig. S3B).”

Line 141-142: “The RNA levels synthesized in the bacteria were consistent with the AND gate profile that was observed in the worm experiment (Fig. S4C).”

Line 157-158: “Similar to the AND gate, the RNA levels measured by RT-qPCR was consistent with the OR gate profile (Fig. S4D).”

Line 371-385: “RT-qPCR Synthesized single-stranded *gfp* RNA was also evaluated by RT-qPCR. Indicated bacteria was first allowed to grow overnight at 37°C with 250 rpm shaking in LB medium with appropriate antibiotics. The overnight culture was then diluted 100 times the next day with supplementation of the appropriate inducers and antibiotics. The total RNA was extracted with RNeasy Mini Kit (Qiagen) when the O.D. reached 0.6-0.8. A total amount of 20 ng of the extracted RNA was then used for RT-qPCR by qScript 1-Step SYBR Green qRT-PCR Kit (Quantabio) on a LightCycler 96 (Roche). Each sample was measured with 3 biological replicates, and each biological replicate had 3 technical replicates on the 96-well plate. The primers used to quantify *gfp* RNA synthesis levels were *gfp*-F: 5'-CGGAGAAGAACTTTTCACTGG-3', and *gfp*-R: 5'-GGTAAGTTTTCCGTATGTTGC-3'. Synthesized *gfp* RNA levels were normalized to the expression of *cysG*⁴ with the primers *cysG*-F: 5'-GATCGCGACTGTCTGATTG-3', and *cysG*-R: 5'-CGGTGAACTGTGGAATAAACG-3'.”

Line 616-618: “(B) RT-qPCR results showed the fold change of the bacterial 200bp single-stranded *gfp* RNA synthesis at indicated IPTG concentrations relative to the control group with no IPTG.”

Line 629-631: “(C) AND-gate and (D) OR-gate controlled 200bp single-stranded *gfp* RNA synthesis levels relative to the control group with no IPTG. RT-qPCR was used to show the *gfp* RNA levels.”

The same questions apply to the protein secreting strain. How much *unc-22* polypeptide is made, is it released from the cell into the media or stored intracellularly?

- Thank you for the question about *unc-22* polypeptide. It is our writing that leads to the confusion. We didn't produce *unc-22* polypeptides using bacteria. Instead, we produced *unc-22* RNA in the bacteria. When the *unc-22* RNA producing bacteria was ground open, the *unc-22* RNA was released from the bacteria to *C. elegans* and *C. elegans* can absorb the *unc-22* RNA to trigger RNAi in *C. elegans* for twitching behavior modulation.

To address this concern, we have changed line 167-172 from “First, we programmed the twitching behavior in *C. elegans* by interfering with the expression of the *unc-22* protein, which encodes a polypeptide involved in locomotion⁵.” to “First, we programmed the twitching behavior in *C. elegans* by feeding *unc-22* RNA producing bacteria to worm to interfere with the UNC-22 protein expression through RNAi in the worm. Since *unc-22* encodes a polypeptide involved in locomotion⁵, we were able to program the twitching behavior in *C. elegans*.”

More description of the choice of *E. coli* strain needs to be provided in the methods section. Is this a strain found in *C. elegans* or a laboratory strain of *E. coli*? Why was it chosen?

- Thank you for pointing out the missing information on the choice of *E. coli* strain. A wide range of bacteria have been utilized including HT115(*rnc*-), BL21(DE3), and OP50(*xu363*)⁶. HT115 is often chosen for producing dsRNA and triggering RNAi in *C. elegans* because it lacks *rnc* gene, which encodes RNase III and therefore can produce more dsRNA for higher RNAi efficiency in the worm. While the *rnc*- HT115 produces stronger phenotypes, BL21(DE3) shows less RNAi efficiency compared with HT115³. In our specific case, BL21 serves as a better bacteria host because we have been struggling with a complete “on” and “off” status of the RNAi, and BL21(DE3) potentially produces less RNA to achieve a complete “off” state. So we used BL21(DE3) throughout our paper.

To address this question, we have added the following parts in line 74-80. “While many *E. coli* strains including HT115(*rnc*-), BL21(DE3), and OP50(*xu363*) have been utilized to produce RNA for RNAi in worm^{3,6}, within which HT115(*rnc*-) lacks RNase III to

produce most dsRNA³ and BL21 has the lowest RNA producing levels³. We started with BL21 because the relatively lower dsRNA production is correlated with less RNAi efficiency in *C. elegans* and thus can potentially provide a dynamic range of RNAi for a complete “on” and “off” state³. We used BL21(DE3) throughout this work.”

The authors claim they are feeding the *C. elegans* with this bacteria. It is not clear that the bacteria is being ingested by *C. elegans* from the media, or that the delivery of its products is occurring through this mechanism. The authors need to demonstrate how the effectors (RNA or protein) are being delivered to *C. elegans*. Is this occurring through the ingestion of the bacterial cells and activation from within the GI tract or are the molecules produced by the engineered cells being ingested or absorbed from the media by *C. elegans*? These are important questions to help better understand the relevance of this work for other orally delivered engineered microbes.

- Thank you for the questions. As mentioned in the answer to the first question, inhibitory RNA is delivered from the bacterial cell when *C. elegans* grind on the bacteria by pharynx and absorb the bacterial contents. The absorbed dsRNA will then be distributed throughout the body leading to the silence of targeted genes in areas that are not initially exposed to the dsRNA. (same as the first question)

Other developers of synthetically engineered microbes have surpassed the accomplishment of this work by demonstrating *in vivo* expression of engineered genes by the bacterial cells in mammalian systems including humans. These additional studies should be referenced.

- Thank you for reminding us to include references about synthetically engineered microbiomes with *in vivo* bacterial expression genes in the mammalian system. We have made changes by adding the following references into the manuscripts line 37-40. “And engineering bacteria have demonstrated their potentials with pest control^{7,8}, plant growth promotion⁹, human disease diagnosis¹⁰⁻¹², and therapeutics¹³. However, little has been done in terms of transferring the decision-making logics from bacteria to the animals to program animal physiology and behaviors.”

The AND/OR system in itself is not novel and it would have been more relevant if the chosen promoter inducer systems related to effectors that could only be produced, or conditions that could only turn on inside the worm. More discussion about how such AND/OR systems could be beneficial, rather than just simply turning on the gene in the host when it is needed, such as with a temperature or anaerobic promoter should be added to the discussion.

- We appreciate your suggestions. One powerful benefit of using logic gates in the microbiome engineering aspect is the precise control over the bacteria for host physiology and behavior modulation, e. g. turning on the target genes expression in a positional and signal specific manner responding to positional signals and toxicity/nutrients/disease biomarkers.

In this manuscript, we demonstrated programming host animal through engineered bacteria using AND/OR gates responsive to defined chemicals (IPTG, aTc, Ara). For ongoing projects in the lab, we are engineering our bacteria to sense new and more practical signals including nutrients (fructose, glucose, or short-chain fatty acids), disease biomarkers (cancer markers, hemes, inflammation molecules), gut positional specific signals (pH, oxygen, short-chain fatty acid concentration), orthogonal signals (lights, temperature, and orthogonal chemicals) to make the programming more applicable for real-life functions. So that, we can use these logic gates to program metabolites production for host physiology and behavior programming.

To address this question, we have added the following sentences to the discussion section in line 248-253. “We could extend our programming with *in situ* signal detection and responses for host physiology and behavior modulations by further engineering bacteria to sense the host’s native signals, usually in the form of nutrients (sugar^{14,15} and hormones^{16,17}), disease biomarkers (cancer markers¹⁸, hemes¹⁹, inflammation molecules^{10,11}), gut positional specific signals (pH^{20,21}, oxygen^{22,23}, short-chain fatty acid concentration²⁴), and orthogonal signals (lights²⁵, temperature²⁶, and orthogonal chemicals²²).”

Based on these comments I would suggest the manuscript needs significant revision before resubmitting to Nature Communications for Review.

Some additional minor points on the manuscript are below:

- Please provide more detail in the abstract about which genes are being modulated in *C. elegans*

- Thank you for the suggestions, we have made changes in the abstract line 10-13.

“Here, we present a bacteria-animal symbiont system in which engineered bacteria recognize external signals and modulate animal gene expression, twitching behavior, and fat metabolism through RNA interference toward *gfp*, *sbp-1*, and *unc-22* gene in *C. elegans*.”

- Line 32, a word is missing in this sentence. “...because it will yield a greater understanding than we have so far of how biological phenomena...”

- Thank you for the suggestions, we have made changes accordingly to “Nevertheless, such programming is worthwhile because it will yield a greater understanding than we have so far of how biological phenomena emerge from genetic and cellular events.”

- Figure 2B,C: what concentration of IPTG was used for this study? Why is there no impact of IPTG in 2B but loss of GFP in 2C?

- Thank you for the suggestions, 1 mM of IPTG was used in this set of experiments. Figure 2C represents the ss200 column from the quantification results in 2B.

We have made the following changes in line 118-120 from “Quantification of *C. elegans* GFP fluorescence intensity controlled by bacteria yielded single-stranded GFP RNA of various lengths” to

“Quantification of *C. elegans* GFP fluorescence intensity controlled by bacteria yielded single-stranded *gfp* RNA of various lengths with no IPTG and and 1mM IPTG.”

We have also changed line 120-122 from “Optical and fluorescent images of *C. elegans* SD1084 GFP expression programmed by bacterial single-stranded 200 bp GFP RNA” to

“Optical and fluorescent images of *C. elegans* SD1084 GFP expression programmed by bacterial single-stranded 200 bp *gfp* RNA with no IPTG and 1mM IPTG.”

- The relevance of the size of the RNA fragment should be discussed

- Thank you for the suggestions. -Size of the bacterial synthesized RNA affects RNAi efficiency. Shorter RNA has a lower binding affinity toward SID-1 protein that is responsible for transporting the RNA into *C. elegans* cells.

We have made the following changes in line 91-95. “Systemic RNAi in *C. elegans* requires systemic RNA interference deficiency-1 (SID-1) protein for the transport of RNA^{5,27}. SID-1 has been shown to have a lower binding affinity for shorter RNA sequences²⁷ leading to lower RNAi efficiency. We hypothesized that we could achieve the right dynamic range of RNAi by reducing the length of the RNA and thus less transporting RNA into the host with less RNAi efficiency.”

- Figure 3 and supplemental figures: In Figure 3B the output table for the OR circuit should this show a “1” when either L-Arabinose or Anhydrotetracycline are present? Check also the supplemental figures for similar error.

- Thank you for the suggestions, we have made the corrections in Figure 3B, Figure S4A, and Figure S4B.

- Figure 4. The authors need to describe how many worms were counted in these studies. How many are enough to conclude 100% twitching?

- Thank you for the suggestions, we have added the following information in line 340-346 for more accurate statistical information. “For *gfp* and *sbp-1* RNAi experiments, 5 worms from each condition were analyzed in one experiment, and each experiment was repeated 3 times (n=15 worms from each condition were analyzed in total)^{28,29}. For the twitching behavior, over 200 worms were analyzed from each condition. In each condition, we observed either twitching in all worms or no twitching at all. RNAi triggered 100% twitching behavior was also observed in other study³. The number of worms we used for calculation is in the same range as other paper when dealing with GFP quantification, fat storage, and twitching behaviors^{28,29}.”

- We have also redone the statistical test and made changes as follows:

Line 163-164: “(***P < 0.001 t-test with Bonferroni correction. Error bars show the mean \pm standard deviation for n=15)”

Line 208-209: “(***P < 0.001 t-test with Bonferroni correction. Error bars show the mean \pm standard deviation for n=15)”

Reviewer #2 (Remarks to the Author):

Gao and Sun describe genetic circuits that can induce RNAi in *C. elegans* by feeding bacteria. They engineer "AND" and "OR" logic gates into bacteria that synthesize single or double-stranded RNA, which are fed to worms to trigger RNAi. The bacterial genetic circuits utilize split or tandem T7 polymerases that conditionally activate T7 transcription in response to single or dual inducers (IPTG, Arabinose, and Tetracycline) to generate the AND and OR gates, respectively. The authors use this system to downregulate a GFP reporter, *unc-22*, and *sbp-1*. The authors claim that this represents an original example of programming *C. elegans* physiology, behavior, and metabolism through ingested bacteria. They also assert that their system explores the widely unexplored bio-computational design of higher organism, leading to metabolic and behavioral programming for agricultural, therapeutic, and basic science applications.

Dear reviewer, thank you for your comments concerning our manuscript entitled “Programming animal physiology and behaviors through engineered bacteria”. These comments help us improve our manuscript and provide important guiding significance to our research. We have studied your comments thoroughly and made corrections/responses to address your concerns.

Major concerns

Genetic circuits are promising synthetic biology tools that have great potential in resolving the complexity of engineering and programming biological systems. However, their prior implementations have mostly been limited to bacteria and mammalian cell cultures. In principle, Gao and Sun present an elegant and novel approach to program higher multicellular organisms by exploiting their innate silencing mechanism and their relationship to bacterial symbionts. Programming a simple multicellular organism such as *C. elegans* by feeding bacteria would potentially be of great interest in the synthetic biology field.

However, the examples that the authors develop are, unfortunately, trivial. If one accepts two well-established premises: (1) RNAi by feeding can induce phenotypes (e.g., Timmons and Fire, 1998 *Nature* and Fraser et al., 2000, *Nature*) and (2) bacteria can be engineered to express RNA in response to one or more inducers, then the authors merely recapitulate RNAi phenotypes that have been widely observed in hundreds of laboratories in response to RNAi by reduced, animals twitch in response to

levamisole, or animals deposit more fat, respectively. So, in my view, the manuscript does not describe any new results or represent a true innovation. For the manuscript to represent a real advance, I would expect the authors to engineer a new functionality into the organism and then control this functionality, e.g., with bacteria that express RNA in response to one or more inducers.

- We appreciate your inputs here. We do recognize that 1) RNAi is well studied, and 2) bacteria has been engineered with genetic circuits for targeted RNA/protein/metabolites expression. However, our motivation is based on the hypothesis that we can transfer the genetic circuits from bacteria to animals via the bacteria-animal symbiotic relationship. Compared with injecting RNA directly (which is well studied), bacteria can sense the environment, make decisions, and produce the RNA for decision transfer purposes. RNA happens to be the communication intermediates we picked between the bacteria and the model animal *C. elegans*. Our argument is that adapting a well-studied system is a good start for this research direction.

In fact, we have two major directions for other projects going on in our lab. First, we are using *C. elegans* and bacteria as a high-throughput screening platform to identify key bacterial metabolites modulating animal physiology and behaviors. These metabolites will engineer a new functionality into the host with mechanisms identified. Secondly, we are engineering our bacteria to sense new and more practical signals including nutrients (fructose, glucose, or short-chain fatty acids, et al.), gut positional specific signals (pH, oxygen, short-chain fatty acid concentration), or orthogonal signals (lights and new chemicals) to make the programming more practical for real-life functions. With all the projects going on, we can use these logic gates to control the identified metabolites production for more host physiology and behavior programming. Although these projects are still under development in our lab and are likely to take more time to finish, we believe this manuscript would serve as a stepstone to further enhance this project.

In summary, we agree with you that RNAi is a well-studied intermediate but hope that we have convinced you that this is the very first step showing logic gates transfer from bacteria to animals for future innovations.

In my view, the authors also greatly overstate the importance of the advance. I do not believe that inducing simple phenotypes constitute "programming the physiology and behavior" of *C. elegans*. I am also not sure how the authors' system "can be developed for applications to pest control, disease diagnostics, and the delivery of therapeutics"?

- Thank you for the comments. We did forget to put in details while mentioning that our system can be developed for pest control, disease diagnostics, and therapeutics delivery. Our system can be utilized for pest control because insects and worm related pest can be engineered through environmental bacteria and RNA interference^{7,30}. In our case, we can deliver bacteria to kill insects with environmental signals, which can specifically target insects while being environmentally friendly. Meanwhile, by engineering bacteria with genetic circuits that detect disease biomarkers and delivers

therapeutics including colon cancer therapeutics, we can knock in disease diagnostics and therapeutics functions from bacteria to modulate higher organisms.

To address this question, we have added the following parts to the discussion section in line 249-254. “We could extend our programming with *in situ* signal detection and responses for host physiology and behavior modulations by further engineering bacteria to sense the host’s native signals, usually in the form of nutrients (sugar^{14,15} and hormones^{16,17}), disease biomarkers (cancer markers¹⁸, hemes¹⁹, inflammation molecules^{10,11}), gut positional specific signals (pH^{20,21}, oxygen^{22,23}, short-chain fatty acid concentration²⁴), and orthogonal signals (lights²⁵, temperature²⁶, and orthogonal chemicals²²).”

We have also changed line 259-264 from “Our system can be developed for applications to pest control, disease diagnosis, and the delivery of therapeutics.” to “By fully exploring our platform, we could potentially reprogram insects and worm physiology and behaviors through bacterial RNA for RNAi toward agriculture applications. Meanwhile, by engineering bacteria with logic gates to recognize nutrients, disease biomarkers, gut positional signals, and orthogonal chemicals, we can potentially use our system for disease diagnosis and therapeutics delivery for mammals including human beings.”

General comments

From a more general perspective, I am not sure that RNAi by feeding is a particularly good choice for inducing phenotypes in response to external stimuli. The response times are slow and are poorly multiplexed with only partial effects when two genes are silenced and only rare effects when three genes are targeted. Importantly, genes expressed in neurons cannot be easily targeted (although this can be solved by engineering, see e.g., Firnhaber & Hammarlund 2013 PLoS Genetics) which would be the main tissue to for engineering "behavior". In the examples demonstrated by the authors, the output comes from disrupting genes during development. For most interesting behaviors, it would be desirable to rapidly induce, e.g., protein dimerization or protein degradation in response to light or chemical cues to change the behavior of a fully developed animal. These tools are widely available in other organisms and have also been used successfully in *C. elegans* (see e.g. Fielmich et al. 2018 Elife).

- Thank you for your comments and suggestions. Genetic engineering on animal directly with signal responsive proteins are much faster like the example you mentioned. However, it needs direct engineering on the animal, which needs mammalian level precise genetic circuit design, efficient DNA circuits delivery, and ethics efforts. The motivation for our work is to program animal physiology and behavior without genetically modifying the animal. Utilizing microbe-animal relationships to modulate and program animal is relatively slower (hours based)^{31,32}, but has the advantage of being much easier to manipulate and deliver (without genetic engineering on the animal directly). We don't think RNAi by feeding is the best choice for many other applications. For example, for mammals, metabolites/neuron transmitters are a better choice compared

with RNAi. The potential approach to make the process faster is to use enzyme dimerization or protein degradation for metabolites production, which will be a much faster process than promoter induced transcription and translation.

We have also talked about the other current projects in the lab from the earlier sections. Hopefully, we have convinced you that we are on the right track.

The authors argue that the use of short RNAs are necessary to achieve an appropriate dynamic range and that they “tune” the responses. But these observations almost certainly have to do with their initial choice of a leaky lac promoter. All the responses that the authors describe are “all or none” and do not involve any “tuning” of responses. In the absence of RNA expression from bacteria then animals will remain fluorescent, will not twitch, or deposit extra lipids. Once a threshold of RNA is produced, then the secondary small RNA amplification system (22G-RNAs, primarily) will kick in and silence the transgene or endogenous genes.

- Thank you for the comments. We recognize that the lac promoter was the problem in the beginning and using shorter RNAs is not the only solution.

Our point is that even if we started with a leaky promoter, we were still able to get to the right dynamic range for the ON/OFF status. There is no guarantee that promoters we are going to use in the future will come with a good dynamic range. With a new promoter that doesn't have a good dynamic range, one of the solutions is to optimize on the promoters. And using shorter RNA for the assay is one promising approach to solve these challenges. It's not the only solution but works well for GFP silencing and fat storage modulation in the manuscript.

For example, we can work on the promoter dynamic range to achieve a complete “off” and “on” status of RNAi. However, in reality, we don't have much freedom picking promoters with proper dynamic ranges because most of the time, we usually use promoters based on the inducers (target molecules). Different promoters will have different dynamic ranges unless optimization “tuning” was done.

Despite my reservations about the current version of the manuscript, I do hope that the authors will persevere and further develop their system to truly engineer worms to control behaviors (very few would consider muscle spasms in response to a drug a “true” behavior). The presented results represent a good set of validation experiments that confirm their ability to reproduce previously published phenotypes. And the results would set the stage for truly engineering the behavior or physiology of a simple multicellular organism.

Specific comments and suggestions

The authors used a curious strain for the GFP experiments. SD1084 has a sur-5:GFP plasmid that expresses GFP in all cells but also has a flag-tagged GES-1 (a carboxylesterase) protein overexpressed. The dynamic range of the GFP silencing is

likely reduced because neuronal expression will not be silenced (which is likely why there is a 20% residual fluorescence). The strain background would not be expected to interfere with the responses but the authors may want to comment on this in the manuscript.

- Thanks for the comment, and we agree that neuronal expression is difficult to target, sometimes it needs genetic modification on the worm to achieve efficient neuronal gene targeting³³. So, it is possible that the 20% residual fluorescence is from the neuronal expression that is difficult to target.

We have added the following information in line 110-114 to make it clearer. "What is worth mentioning here is that the *C. elegans* strain SD1084 we picked express nuclear-localized SUR-5::GFP²⁹ in all cells with a flag-tagged GES-1 (a carboxylesterase) protein co-overexpressed²⁹. Since neuronal genes are more difficult to target with RNAi³³, we usually see around 20% residual fluorescence on our "off" stage, which should come from the neuronal GFP that were not RNAi silenced."

Lines 82-86 and Figure 2. It is curious, and unexpected, that single-strand RNA leads to silencing. Is the single-stranded DNA in fact two inverted repeats of GFP so a dumb-bell formed dsRNA is formed (see Timmons et al. 2001 for a configuration that works in this way)? If the fragment is only one strand of GFP then the figure labeling should be changed from dsGFP to ssGFP.

- We were also surprised that ssRNA leads to silencing. We started by using ssRNA as a negative control but later found that they also silenced GFP in the worm. And we found in the literature that ssRNA also worked for RNAi³⁴. In our experiment setup, we did not have a dumb-bell structure.

To address your comments, we have changed our Figure 2 labeling to "(B) Quantification of *C. elegans* GFP fluorescence intensity controlled by bacteria yielded single-stranded *gfp* RNA of various lengths with no IPTG and 1 mM IPTG. (C) Optical and fluorescent images of *C. elegans* SD1084 GFP expression programmed by bacterial single-stranded 200 bp *gfp* RNA with no IPTG and 1mM IPTG."

Also, the choice of RNAi bacteria is unusual. Almost all RNAi by feeding experiments are done using the HT115 strain that lacks a double-strand RNase activity (see same paper by Timmons et al.). The authors are likely to get a better dynamic range and more robust effects if using they switch to the canonical strain, unless HT115 cannot be engineered?

- Thank you for pointing out the missing information on the choice of *E. coli* strain. A wide range of bacteria have been utilized including HT115(*rnc*-), BL21(DE3), and OP50(*xu363*)⁶. HT115 is often chosen for producing dsRNA and triggering RNAi in *C. elegans* because it lacks *rnc* gene, which encodes RNase III and therefore can produce more dsRNA for higher RNAi efficiency in the worm. While the HT115(*rnc*-) produces

stronger phenotypes, BL21(DE3) shows less RNAi efficiency compared with HT115³. In our specific case, BL21 serves as a better bacteria host because we have been struggling with a complete “on” and “off” status of the RNAi and BL21(DE3) potentially produces less RNA to achieve a complete “off” state. So, we used BL21(DE3) throughout our paper.

To address this question, we have added the following parts in line 74-80. “While many *E. coli* strains including HT115(*rnc*-), BL21(DE3), and OP50(*xu363*) have been utilized to produce RNA for RNAi in worm^{3,6}, within which HT115(*rnc*-) lacks RNase III to produce most dsRNA³ and BL21 has the lowest RNA producing levels³. We started with BL21 because the relatively lower dsRNA production is correlated with less RNAi efficiency in *C. elegans* and thus can potentially provide a dynamic range of RNAi for a complete “on” and “off” state³. We used BL21(DE3) throughout this work.”

Please rephrase the sentence spanning lines 111 - 113. This sentence is confusing and inaccurate. The bacteria do not transcribe GFP (they make a fragment of *gfp*, and it's unclear if it is single stranded or double-stranded and if it is in the coding vs reverse direction). Also, RNAi is a process, so “RNAi expression” is not triggered in the worms.

- Thank you for pointing it out. We have changed the sentence (now lines 136-139) to: “Both L-arabinose (Ara) and anhydrotetracycline (aTc), the respective inducers of these promoters, are needed to induce synthesis of *gfp* ssRNA coding strand fragment in the bacteria. The *gfp* RNA fragment will then lead to RNAi in the worm after the worm grinds the bacteria open and absorb the RNA fragment.”

We have also changed all the gene names to italic and protein name to capital as you suggested. Thank you!

Line 72-74: “To achieve that, we put *gfp* dsRNA synthesis under the control of a bidirectional lac promoter, which induces the synthesis of RNA in *E. coli* under IPTG induction.”

Others changes include line 80-81, line 83-86, line 99-100, line 128-129, line 161-162, line 152-155, line 171-173, line 179-181, line 182-184, line 187-189, line 601-605, line 610-616, and line 293-295.

Statistics: It would be helpful if the authors indicate the number of experiments performed for each statistical comparison and whether graphs so standard deviations or standard error of means. Also, the statistical test (Student's T-test) used in figure 3 and 4 does not appear to have been corrected for multiple comparisons. The differences are robust and should easily hold up.

- Thank you for the suggestions, we have added the following information in line 340-346 for more accurate information. "For *gfp* and *sbp-1* RNAi experiments, 5 worms from each condition were analyzed in one experiment, and each experiment was repeated 3 times (n=15 worms from each condition were analyzed in total)^{30,39}. For the twitching behavior, over 200 worms were analyzed from each condition. In each condition, we observed either twitching in all worms or no twitching at all. RNAi triggered 100% twitching behavior was also observed in other study³. The number of worms we used for calculation is in the same range as other paper when dealing with GFP quantification, fat storage, and twitching behaviors.^{30,39}"

-We have also redone the statistical test and made changes as follows:

Line 163-164: "(***P < 0.001 t-test with Bonferroni correction. Error bars show the mean ± standard deviation for n=15)"

Line 208-209: "(***P < 0.001 t-test with Bonferroni correction. Error bars show the mean ± standard deviation for n=15)"

Reviewer #3 (Remarks to the Author):

In this work, the authors designed and engineered logic circuits in the model bacterium *Escherichia coli* to study programmable outputs in the physiology and behaviour of its host, *Caenorhabditis elegans*. The scope of this study is interesting and potentially a good foundational proof-of-concept study for using engineered bacteria to predict and program animal physiology for specific applications. However, the overall data are insufficient and do not fully support the conclusions of this study. The following comments should be addressed before the manuscript can be re-considered for publication.

Major Comments

1. The introduction section is inadequate in highlighting the background and importance of RNA in host-microbe interaction. Please highlight the importance more clearly to justify the direction and scope of this study.

- Thanks for the comment. We have made changes as follows to highlight the background and importance of our work:

Line 32-34: "Nevertheless, such programming is worthwhile because it will yield a greater understanding than we have so far of how biological phenomena emerge from genetic and cellular events."

Line 37-40: "And engineering bacteria have demonstrated their potentials with pest control^{7,8}, plant growth promotion⁹, human disease diagnosis¹⁰⁻¹², and therapeutics¹³. However, little has been done in terms of transferring the decision-making logics from bacteria to the animals to program animal physiology and behaviors."

Line 249-254: “We could extend our programming with *in situ* signal detection and responses for host physiology and behavior modulations by further engineering bacteria to sense the host’s native signals, usually in the form of nutrients (sugar^{40,41} and hormones^{52,53}), disease biomarkers (cancer markers⁵⁴, hemes⁵⁵, inflammation molecules^{21,22}), gut positional specific signals (pH^{56,57}, oxygen^{58,59}, short-chain fatty acid concentration⁶⁰), orthogonal signals (lights², temperature⁶¹, and orthogonal chemicals⁵⁸)”

We have also changed line 259-264 to “By fully exploring our platform, we could potentially reprogram insects and worm physiology and behaviors through bacterial RNA for RNAi toward agriculture applications. Meanwhile, by engineering bacteria with logic gates to recognize nutrients, disease biomarkers, gut positional signals, and orthogonal chemicals, we can potentially use our system for disease diagnosis and therapeutics delivery for mammals including human beings.”

2. The experimental rationale for RNA circuit design in *E. coli* is unclear and the following points should be addressed.

- Additional information on initial dsGFP RNA tested in figure S1 should be provided as this was not described sufficiently.

- Thank you for the comments. We have made the following changes in line 593-597 “A 750 bp fragment for *gfp* from *C. elegans* SD1084 was inserted between two T7/LacO that are in reciprocal orientation on a pET24a plasmid vector. The plasmid was then transformed into *E. coli* BL21(DE3) and fed to SD1084 worms cultured on NGM plates with or without 1mM IPTG. The *E. coli* BL21(DE3) without any plasmids was also fed to SD1084 worms as a negative control.”

- In figure 1 and S2; it was hypothesized that the initial circuit did not produce the desired results as the lac promoter was leaky. This aspect was not addressed and subsequently, the same promoter was employed in downstream circuitry design as shown in figure 1 and S2. Please address the possibility, where promoter leakiness is the reason for the result obtained in Figure S1, and provide relevant data that either support or reject such a possibility.

- We had the hypothesis that lac promoter was leaky because research demonstrated that RNAi efficiency is correlating with the RNA concentration³⁵ and T7-lac promoter was known for leaky expression in various systems³⁶⁻³⁸.

For results that we tried quick and dirty (by observation only), we used continuous promoter (ProD) to regulate T7 lysozyme expression, which can reduce the T7-lac promoter leakiness by specifically binding with the T7 polymerase^{39,40}. We observed reduced RNAi efficiency in a small fraction of the worms, which indicates reducing the leaky expression of dsGFP RNA will help reduce the RNAi a little bit. Also, according to the literature, the efficiency of the RNAi increased with the amount of RNA produced and fed number of the bacteria (RNA)³. The amount of synthesized RNA can be quantified by quantitative reverse transcription PCR (RT-qPCR). We also added the RT-qPCR results indicating the bacterial RNA synthesis in the manuscript in Fig. S3B, Fig.

S4C, and Fig. S4D. Although in the end we still need to screen out different length RNA for a clear “ON” and “OFF” status of the RNAi, in the following experiments, we used ProD for all the genetic circuits to reduce the background RNA expression.

To better explain it, we have added the following information in line 298-302.

“We have added ProD promoter into the plasmid to overexpress promoter repressors, which improves the tightness of the pTet and pBAD promoters in the genetic circuit. We also used ProD to produce T7 lysozyme to reduce the background synthesis of RNA from lac promoter by binding with the T7 lysozyme to reduce the background transcription of RNA product^{39,40}.”

For your second question about “subsequently, the same promoter was employed in downstream circuitry design as shown in figure 1 and S2.”

Although we kept the same lac promoter for our circuit design, we screened out shorter RNAs, which helped reduce the RNAi efficiency. Shorter RNA has a lower binding affinity with the SID-1 protein that is responsible for transporting the RNA into *C. elegans* cells and our experiments demonstrated that adapting shorter single-stranded or double-stranded RNA helped reduce the RNAi efficiency in animals.

- The authors’ choice of reducing RNA length seems contradictory to the function of SID-1. Please explain clearly the link between SID-1’s affinity for longer RNA and the choice for shorter RNA lengths.

- Thank you for the suggestions. The size of the bacterial synthesized RNA affects RNAi efficiency. Shorter RNA has a lower binding affinity with the SID-1 protein that is responsible for transporting the RNA into *C. elegans* cells.

We have made the following changes in lines 91-95 to make it easier to read and understand. “Systemic RNAi in *C. elegans* requires systemic RNA interference deficiency-1 (SID-1) protein for the transport of RNA^{5,27}. SID-1 has been shown to have a lower binding affinity for shorter RNA sequences²⁷ leading to lower RNAi efficiency. We hypothesized that we could achieve the right dynamic range of RNAi by reducing the length of the RNA and thus transporting less RNA into the host with less RNAi efficiency.”

3. In figure 3B; The OR gate output is inaccurate. The logical expression of OR gate in this context should be:

L-Ara aTC Output

-- 0
- + 1
+ - 1
++ 1

where 0 = no GFP silencing and 1 = GFP silencing. This is also observed in Figure S4A and S5A. Please correct the OR gate outputs in these figures accordingly.

- Thank you for the suggestions. We have made the corrections in Figure 3B, Figure S4A, and Figure S5A.

4. Please explain and justify the use of promoter ProD in the circuit design.

- Promoter ProD is a constitutive promoter that we use to produce repressors for promoters⁴¹ and T7 lysozyme³⁹, which reduce the background expression level of lac, pTet, and pBAD promoter.

To better explain it, we have added the following information in line 298-302.

“We have added ProD promoter into the plasmid to overexpress promoter repressors, which improves the tightness of the pTet and pBAD promoters in the genetic circuit. We also used ProD to produce T7 lysozyme to reduce the background synthesis of RNA from lac promoter by binding with the T7 lysozyme to reduce the background transcription of RNA product^{39,40}.”

5. Optimization data were used to establish the combination of inducer concentrations to demonstrate OR and AND gate is not provided. In the same context, the arabinose concentrations used for AND gate and OR gate is significantly different. Please provide the necessary data and justify the inducer concentrations used in this assay.

- We have added the *unc-22* and *sbp-1* RNA length and inducer concentration information in the manuscript.

Line 204-206: “*C. elegans* twitching behavior in response to OR and AND gate programming through bacteria expressing 200 bp and 400 bp double-stranded *unc-22* RNA.”

We have also added the data in Figure S5 and S6 showing the concentration profiles of *unc-22* and *sbp-1* results.

Other inducer concentrations were already available in the materials and methods section in lines 328-333.

6. In figure 4B; the final *unc-22* RNA used for figure 4B is not specified along with screening data for *unc-22* RNAi candidates. Please provide necessary data to support the results in figure 4B with a detailed explanation for screening double-stranded RNA instead of single-stranded RNA for *unc-22*. This approach is contrary to earlier results shown in figure 1 and 2 where single-stranded RNA was used to establish circuit design.

- RNAi targeting at different proteins have different efficiency. In our case, GFP RNAi works better with single-stranded RNA, SBP-1 RNAi works fine with full length double-stranded RNA, and UNC-22 RNAi works at shorter double-stranded (400bp and 200bp) RNA. So, there is no universal rules about double-stranded or single-stranded or the length of the RNA we should use for different target.

We have added the *unc-22* RNA length information in the manuscript figure 4B description.

Line 204-206: "*C. elegans* twitching behavior in response to AND and OR gate programming through bacteria expressing 400 bp and 200 bp *unc-22* RNA."

We have also added the different length *unc-22* RNA screening and inducer concentration results in Figure S5.

Line 643-667: "Figure S5. Screening on *unc-22* dsRNA/ssRNA length and inducer concentrations. (A) We used 1 mM IPTG and no IPTG to induce different length *unc-22* dsRNA and ssRNA under lac promoter in bacteria BL21(DE3) and fed them to worms. We found that both 200 bp and 400 bp *unc-22* dsRNA and ssRNA led to complete "on" and "off" status of worm twitching behaviors. (B) We constructed 200bp *unc-22* dsRNA and ssRNA under AND gate genetic circuits and inducer concentrations of 0.2 mg/mL of L-arabinose and 0.1 μ g/mL of anhydrotetracycline. After feeding these bacteria to worms, we found that only a small fraction of the worms showed twitching behavior. (C) Same procedure was conducted on 400bp *unc-22* dsRNA and ssRNA. 400bp dsRNA induced 100% twitching behaviors in worms in presence of both inducers but the arabinose control also lead to some level of worm twitching. 400bp *unc-22* ssRNA achieved 22% of worm twitching behavior with both inducers, which is not sufficient to serve as "on" status. (D) Using 400 bp dsRNA as the platform, we further tried different inducer concentrations. Decreasing inducer L-arabinose concentration to 2 μ g/mL gave a good AND gate profile. Numbers on the x-axis indicate the inducer concentrations in μ g/mL. (Error bars show the mean \pm standard deviation for n=15)."

7. In figure 4B; twitching behavior observed shows high variability without quantitative and secondary qualitative measures. Please ensure the reliability and robustness of the data provided for the twitching behaviour observed along with providing error bars for the figure.

- Thank you for the suggestions, we have added the following information in line 342-346 for more accurate information. "For the twitching behavior, over 200 worms were analyzed from each condition. In each condition, we observed either twitching in all worms or no twitching at all. RNAi triggered 100% twitching behavior was also observed in other study³. The number of worms we used for calculation is in the same range as other paper when dealing with GFP quantification, fat storage, and twitching behaviors^{28,29}."

We have also redone the statistical test and made changes as follows:

Line 163-164: "(***P < 0.001 t-test with Bonferroni correction. Error bars show the mean \pm standard deviation for n=15)"

Line 208-209: "(***P < 0.001 t-test with Bonferroni correction. Error bars show the mean \pm standard deviation for n=15)"

8. Screening and validation for *sbp-1* RNAi candidates are lacking similar to *unc-22*.

Please provide the required data.

- Thanks for the comments. We didn't do screening for *sbp-1* because full-length *sbp-1* worked without having to screen for different lengths of RNA.

To address this comment, we have added the following information in the manuscript line 188-189. "Since full length *sbp-1* RNA works to modulate fat storage in *C. elegans*, we didn't conduct RNA length experiments as in the *unc-22* experiment."

9. The naming of parts in circuit design is inconsistent in the text leading to difficulties in understanding the circuit. For example, Page 6 lines 10 and 15 mentioned promoters control GFP transcription in engineered bacteria, this should rather be the transcription of single-stranded RNA against GFP in engineered bacteria. Please keep the naming of parts consistent in the text to avoid confusion.

- Thanks for the comments. We have made the following changes.

Line 291-292: "Using this method, pET24a-dsgfp, pET24a-dsgfp-100bp, pET24a-dsgfp-400bp, pET24a-dsunc-22-200bp, and pET24a-dsunc-22-400bp were constructed."

Line 301-302: "Plasmids pET24a-ssgfp-100bp and pET24a-ssgfp-400bp were synthesized this way"

Line 317-318: "To program *C. elegans* GFP expression using IPTG, plasmids pET24a-T7-dsGFP, pET24a -dsgfp-100bp to pET24a-dsgfp-400bp were transformed into *E. coli* BL21(DE3) separately, cultured, induced, and fed to the worms."

Line 320-322: "To program GFP expression with an AND gate, *E. coli* BL21 was co-transformed with pTet-Ara-Split-T7-AND-Gate-proD and pET24a-ssgfp-200bp. For OR gate programming, pTet-Ara-OR-Gate-proD and pET24a-ssgfp-200bp were co-transformed into *E. coli* BL21."

Line 325-330: "For twitching behavior AND gate programming, *E. coli* BL21 was co-transformed with pTet-Ara-Split-T7-AND-Gate-proD and pET24a-dsunc-22-400bp. RNA synthesis was induced by 2 µg/mL L-arabinose and/or 0.1 µg/mL anhydrotetracycline. To program twitching behavior using an OR gate, *E. coli* BL21 was co-transformed with pTet-Ara-OR-Gate-proD and pET24a-dsunc-22-200bp. RNA expression was induced by 0.2 mg/mL L-arabinose and/or 0.1 µg/mL anhydrotetracycline."

10. In figure S4A & S4B; there is an incorrect citation of this figure in the text where figure S4A shows OR gate results while figure S4B shows AND gate results. Please revise accordingly.

- Thank you for the comments. We have corrected sentences in line 136-139 to "With this system, worm GFP expression was silenced by more than 70% in the presence of both signals (Ara and aTc) whereas GFP was still expressed (more than 95%) with either no signal or only one of the two signals (Fig. 3A, S4A)."

We have also changed line 147-157 to “We observed that the addition of one or both of the inducers led to more than 90% reduction in *C. elegans* GFP intensity, while the no-inducer control group only had a 24% reduction (Fig. 3B, S4B).”

- 1 Timmons, L. & Fire, A. Specific interference by ingested dsRNA. *Nature* **395**, 854-854, doi:10.1038/27579 (1998).
- 2 Kamath, R. S., Martinez-Campos, M., Zipperlen, P., Fraser, A. G. & Ahringer, J. Effectiveness of specific RNA-mediated interference through ingested double-stranded RNA in *Caenorhabditis elegans*. *Genome Biol* **2**, RESEARCH0002-RESEARCH0002, doi:10.1186/gb-2000-2-1-research0002 (2001).
- 3 Timmons, L., Court, D. L. & Fire, A. Ingestion of bacterially expressed dsRNAs can produce specific and potent genetic interference in *Caenorhabditis elegans*. *Gene* **263**, 103-112, doi:[https://doi.org/10.1016/S0378-1119\(00\)00579-5](https://doi.org/10.1016/S0378-1119(00)00579-5) (2001).
- 4 Zhou, K. *et al.* Novel reference genes for quantifying transcriptional responses of *Escherichia coli* to protein overexpression by quantitative PCR. *BMC Mol Biol* **12**, 18-18, doi:10.1186/1471-2199-12-18 (2011).
- 5 Winston, W. M., Molodowitch, C. & Hunter, C. P. Systemic RNAi in *C. elegans* Requires the Putative Transmembrane Protein SID-1. *Science* **295**, 2456, doi:10.1126/science.1068836 (2002).
- 6 Xiao, R. *et al.* RNAi Interrogation of Dietary Modulation of Development, Metabolism, Behavior, and Aging in *C. elegans*. *Cell Rep* **11**, 1123-1133, doi:10.1016/j.celrep.2015.04.024 (2015).
- 7 Cagliari, D. *et al.* Management of Pest Insects and Plant Diseases by Non-Transformative RNAi. *Frontiers in Plant Science* **10**, 1319 (2019).
- 8 Bento, F. M. M. *et al.* Gene silencing by RNAi via oral delivery of dsRNA by bacteria in the South American tomato pinworm, *Tuta absoluta*. *Pest Management Science* **76**, 287-295, doi:10.1002/ps.5513 (2020).
- 9 Yang, L. *et al.* Promotion of plant growth and in situ degradation of phenol by an engineered *Pseudomonas fluorescens* strain in different contaminated environments. *Soil Biology and Biochemistry* **43**, 915-922, doi:<https://doi.org/10.1016/j.soilbio.2011.01.001> (2011).
- 10 Daeffler, K. N. M. *et al.* Engineering bacterial thiosulfate and tetrathionate sensors for detecting gut inflammation. *Molecular Systems Biology* **13**, 923, doi:10.15252/msb.20167416 (2017).
- 11 Riglar, D. T. *et al.* Engineered bacteria can function in the mammalian gut long-term as live diagnostics of inflammation. *Nature biotechnology* **35**, 653-658, doi:10.1038/nbt.3879 (2017).
- 12 Mimeo, M., Tucker, Alex C., Voigt, Christopher A. & Lu, Timothy K. Programming a Human Commensal Bacterium, *Bacteroides thetaiotaomicron*, to Sense and Respond to Stimuli in the Murine Gut Microbiota. *Cell Systems* **1**, 62-71, doi:<https://doi.org/10.1016/j.cels.2015.06.001> (2015).
- 13 Chowdhury, S. *et al.* Programmable bacteria induce durable tumor regression and systemic antitumor immunity. *Nature Medicine* **25**, 1057-1063, doi:10.1038/s41591-019-0498-z (2019).
- 14 Moser, F. *et al.* Dynamic control of endogenous metabolism with combinatorial logic circuits. *Mol Syst Biol* **14**, e8605, doi:10.15252/msb.20188605 (2018).
- 15 Leveau, J. H. J. & Lindow, S. E. Appetite of an epiphyte: Quantitative monitoring of bacterial sugar consumption in the phyllosphere. *Proceedings of the National Academy of Sciences* **98**, 3446, doi:10.1073/pnas.061629598 (2001).
- 16 Sperandio, V., Torres, A. G., Jarvis, B., Nataro, J. P. & Kaper, J. B. Bacteria–host communication: The language of hormones. *Proceedings of the National Academy of Sciences* **100**, 8951, doi:10.1073/pnas.1537100100 (2003).

- 17 Freestone, P. Communication between Bacteria and Their Hosts. *Scientifica* **2013**, 361073, doi:10.1155/2013/361073 (2013).
- 18 Huh, J. H., Kittleson, J. T., Arkin, A. P. & Anderson, J. C. Modular Design of a Synthetic Payload Delivery Device. *ACS Synthetic Biology* **2**, 418-424, doi:10.1021/sb300107h (2013).
- 19 Stauff, D. L. & Skaar, E. P. The Heme Sensor System of *Staphylococcus aureus*. *Contributions to Microbiology* **16**, 120-135, doi:10.1159/000219376 (2009).
- 20 Yin, X. *et al.* Pgas, a Low-pH-Induced Promoter, as a Tool for Dynamic Control of Gene Expression for Metabolic Engineering of *Aspergillus niger*. *Applied and Environmental Microbiology* **83**, e03222-03216, doi:10.1128/AEM.03222-16 (2017).
- 21 Torres, R., Dorriz, D. & Saviola, B. Induction of the acid inducible lipF promoter is reversibly inhibited in pH ranges of pH 4.2-4.0. *BMC Res Notes* **11**, 284-284, doi:10.1186/s13104-018-3370-1 (2018).
- 22 Chen, Y. *et al.* Tuning the dynamic range of bacterial promoters regulated by ligand-inducible transcription factors. *Nature Communications* **9**, 64, doi:10.1038/s41467-017-02473-5 (2018).
- 23 Hwang, H. J., Kim, J. W., Ju, S. Y., Park, J. H. & Lee, P. C. Application of an oxygen-inducible nar promoter system in metabolic engineering for production of biochemicals in *Escherichia coli*. *Biotechnology and Bioengineering* **114**, 468-473, doi:10.1002/bit.26082 (2017).
- 24 Tobe, T., Nakanishi, N. & Sugimoto, N. Activation of motility by sensing short-chain fatty acids via two steps in a flagellar gene regulatory cascade in enterohemorrhagic *Escherichia coli*. *Infect Immun* **79**, 1016-1024, doi:10.1128/IAI.00927-10 (2011).
- 25 Fernandez-Rodriguez, J., Moser, F., Song, M. & Voigt, C. A. Engineering RGB color vision into *Escherichia coli*. *Nat Chem Biol* **13**, 706-708, doi:10.1038/nchembio.2390 (2017).
- 26 Repoila, F. & Gottesman, S. Temperature Sensing by the *dsrA* Promoter. *Journal of Bacteriology* **185**, 6609, doi:10.1128/JB.185.22.6609-6614.2003 (2003).
- 27 Li, W., Koutmou, K. S., Leahy, D. J. & Li, M. Systemic RNA Interference Deficiency-1 (SID-1) Extracellular Domain Selectively Binds Long Double-stranded RNA and Is Required for RNA Transport by SID-1. *J Biol Chem* **290**, 18904-18913, doi:10.1074/jbc.M115.658864 (2015).
- 28 Nomura, T., Horikawa, M., Shimamura, S., Hashimoto, T. & Sakamoto, K. Fat accumulation in *Caenorhabditis elegans* is mediated by SREBP homolog SBP-1. *Genes Nutr* **5**, 17-27, doi:10.1007/s12263-009-0157-y (2010).
- 29 Lezzerini, M., van de Ven, K., Veerman, M., Brul, S. & Budovskaya, Y. V. Specific RNA Interference in *Caenorhabditis elegans* by Ingested dsRNA Expressed in *Bacillus subtilis*. *PLoS One* **10**, e0124508, doi:10.1371/journal.pone.0124508 (2015).
- 30 Joga, M. R., Zotti, M. J., Smaghe, G. & Christiaens, O. RNAi Efficiency, Systemic Properties, and Novel Delivery Methods for Pest Insect Control: What We Know So Far. *Front Physiol* **7**, 553-553, doi:10.3389/fphys.2016.00553 (2016).
- 31 Uhr, G. T., Dohnalová, L. & Thaiss, C. A. The Dimension of Time in Host-Microbiome Interactions. *mSystems* **4**, e00216-00218, doi:10.1128/mSystems.00216-18 (2019).
- 32 Kaletsky, R. *et al.* *C. elegans* interprets bacterial non-coding RNAs to learn pathogenic avoidance. *Nature* **586**, 445-451, doi:10.1038/s41586-020-2699-5 (2020).
- 33 Calixto, A., Chelur, D., Topalidou, I., Chen, X. & Chalfie, M. Enhanced neuronal RNAi in *C. elegans* using SID-1. *Nat Methods* **7**, 554-559, doi:10.1038/nmeth.1463 (2010).
- 34 Fire, A. *et al.* Potent and specific genetic interference by double-stranded RNA in *Caenorhabditis elegans*. *Nature* **391**, 806-811, doi:10.1038/35888 (1998).
- 35 Timmons, L., Tabara, H., Mello, C. C. & Fire, A. Z. Inducible systemic RNA silencing in *Caenorhabditis elegans*. *Mol Biol Cell* **14**, 2972-2983, doi:10.1091/mbc.e03-01-0858 (2003).
- 36 Kim, S. K., Lee, D.-H., Kim, O. C., Kim, J. F. & Yoon, S. H. Tunable Control of an *Escherichia coli* Expression System for the Overproduction of Membrane Proteins by Titrated Expression of a

- Mutant lac Repressor. *ACS Synthetic Biology* **6**, 1766-1773, doi:10.1021/acssynbio.7b00102 (2017).
- 37 Rosano, G. L. & Ceccarelli, E. A. Recombinant protein expression in *Escherichia coli*: advances and challenges. *Frontiers in microbiology* **5**, 172-172, doi:10.3389/fmicb.2014.00172 (2014).
- 38 Kang, Y., Son, M. S. & Hoang, T. T. One step engineering of T7-expression strains for protein production: increasing the host-range of the T7-expression system. *Protein Expr Purif* **55**, 325-333, doi:10.1016/j.pep.2007.06.014 (2007).
- 39 Moffatt, B. A. & Studier, F. W. T7 lysozyme inhibits transcription by T7 RNA polymerase. *Cell* **49**, 221-227, doi:10.1016/0092-8674(87)90563-0 (1987).
- 40 Studier, F. W. Use of bacteriophage T7 lysozyme to improve an inducible T7 expression system. *Journal of Molecular Biology* **219**, 37-44, doi:[https://doi.org/10.1016/0022-2836\(91\)90855-Z](https://doi.org/10.1016/0022-2836(91)90855-Z) (1991).
- 41 Davis, J. H., Rubin, A. J. & Sauer, R. T. Design, construction and characterization of a set of insulated bacterial promoters. *Nucleic Acids Res* **39**, 1131-1141, doi:10.1093/nar/gkq810 (2011).

REVIEWER COMMENTS

Reviewer #1 (Remarks to the Author):

I have reviewed the revised manuscript and find the authors to have addressed my major concerns and clarified points that were unclear. I would make a minor editorial suggestion on line 37 sentence: "And engineering bacteria have demonstrated their potentials with pest control^{18,19}, plant growth promotion²⁰, human disease diagnosis²¹⁻²³ ³⁸, and therapeutics²⁴ ³⁹", to not start this sentence with "And".

Otherwise I have no further comments or suggestions for this manuscript.

Caroline Kurtz

Reviewer #2 (Remarks to the Author):

Review of the revised manuscript "Programming animal physiology and behaviors through engineered bacteria".

The revised manuscript from Goa and Sun has addressed many of the textual suggestions and is easier to read. However, the authors have not addressed any of my major concerns from the first review of the manuscript ("So, in my view, the manuscript does not describe any new results or represent a true innovation. For the manuscript to represent a real advance, I would expect the authors to engineer a new functionality into the organism and then control this functionality"). They have added only one minor experiment (quantifying dsRNA expression in a supplemental figure). The rebuttal does not address any of my concerns but entices with the promise of the many ongoing projects in the lab that will show the usefulness of the approach in the future. For serious consideration as a stand-alone-publication, I would think that the authors need to show at least one novel result.

As stated in my initial review, I agree that the overall concept is interesting: if bacteria (and RNA expression) can be used as an intermediary to program a multicellular organism then it would be both novel and useful. However, the examples the authors show in the paper remain trivial and the authors do not in any meaningful way demonstrate that they can program behavior. Turning off a fluorescent protein, making an animal shake by knocking down a muscle protein, or increasing the fat content by reducing a transcription factor required for correct lipid metabolism do not constitute behaviors. Furthermore, the experimental conditions (e.g., a poor choice of a gfp reporter with neuronal expression and an unrelated protein moiety attached) suggest to me that the presented results are best treated as pilot experiments that show promise. For example, the presented results would be appropriate for a grant proposal or a Ph.D. student pre-proposal exam.

The text is easier to read but the authors still wildly overstate the implications of their observations. I do not see any coherent argument for the speculation that these advances can be used for agricultural or therapeutic purposes.

In sum, I do not find that the methodologies or the results described can stand on their own as an independent manuscript. The methodology is not novel nor rigorously validated to an extent that it will open up for new experiments in other laboratories. My concerns about using RNAi as a general approach to program behavior due to the slow response times remain. The authors do not program behavior despite their claims. For the manuscript to be of general interest and useful for other laboratories, I would expect that the authors add a considerable amount of experimental results, some of which the authors describe as ongoing in their laboratory. I wish the authors had used the revision

as an occasion to demonstrate the potential of the approach they describe.

Reviewer #3 (Remarks to the Author):

Most of the comments have been addressed sufficiently, except comments #1, 5 and 8.

Comment #1

(1) The introduction section is inadequate in highlighting the background and importance of RNA in host-microbe interaction. Please highlight the importance more clearly to justify the direction and scope of this study.

(2) While the elaboration is well-substantiated with relevant studies, the elaboration does not highlight RNAi as an intermediate in host-microbe interactions. Also, lines 237-240 seems to be an overreach: "this system can be used for disease diagnosis and therapeutics for mammals including human beings." Physiologically, *C. elegans* is not the most relevant system for disease diagnosis and therapeutics delivery in mammals. It would be better to provide a more nuanced elaboration for human application.

Comment #5

(1) The optimization data used to establish the combination of inducer concentrations to demonstrate OR and AND gates are not provided. In the same context, the arabinose concentrations used for AND gate and OR gate are significantly different. Please provide data to justify the inducer concentrations used in this assay.

(2) The caption for the newly revised Figure S5 does not adequately describe the figure.

Comment #8

(1) Screening and validation for *sbp-1* RNAi candidates are lacking, similar to *unc-22*. Please provide the required data.

(2) The reasoning for not conducting RNA length experiments is confusing and not convincing. From the revised manuscript, *unc-22* RNA length screening was done as shown in Figure S5. Are authors suggesting that RNA length experiments for *sbp-1* were not conducted because the dynamic range/difference between 'on' and 'off' states for *sbp-1* was distinguishable? If so, please explain this clearly.

Reviewer #1 (Remarks to the Author):

I have reviewed the revised manuscript and find the authors to have addressed my major concerns and clarified points that were unclear. I would make a minor editorial suggestion on line 37

sentence: "And engineering bacteria have demonstrated their potentials with pest control^{18,19}, plant growth promotion²⁰, human disease diagnosis²¹⁻²³ , and therapeutics²⁴ 39 ", to not start this sentence with "And".

Otherwise I have no further comments or suggestions for this manuscript.

Caroline Kurtz

Dear reviewer, thank you for your comments concerning our manuscript.

We have corrected the sentence in line 37 from "And engineering bacteria have demonstrated their potentials with pest control^{18,19}, plant growth promotion²⁰, human disease diagnosis²¹⁻²³ , and therapeutics²⁴" to "Engineering bacteria have demonstrated their potentials with pest control^{18,19} , plant growth promotion²⁰ , human disease diagnosis²¹⁻²³ , and therapeutics²⁴ "

Reviewer #2 (Remarks to the Author):

Review of the revised manuscript "Programming animal physiology and behaviors through engineered bacteria".

The revised manuscript from Goa and Sun has addressed many of the textual suggestions and is easier to read.

Dear reviewer, thank you for your time and efforts dictated to our manuscript. We appreciate your comments, especially from a pure biology perspective that you have been looking for a new multicellular functionality from our work. However, our manuscript focuses on new control modalities of multicellular organisms that program gene expression and physiology. Our novelty is from synthetic biology perspective, which will inspire new thoughts over how to manipulate multicellular organisms, especially in the case when programming with logic "AND/OR" gates are beneficial. Controlling and programming on multicellular organisms is challenging and our approach presents an exciting first step by enabling towards this goal.

However, the authors have not addressed any of my major concerns from the first review of the manuscript ("So, in my view, the manuscript does not describe any new results or represent a true innovation. For the manuscript to represent a real advance, I would expect the authors to engineer a new functionality into the organism and then control this functionality"). They have added only one minor experiment (quantifying dsRNA expression in a supplemental figure). The rebuttal does not address any of my concerns but entices with the promise of the many ongoing projects in the lab that will show the usefulness of the approach in the future. For serious consideration as a stand-alone-publication, I would think that the authors need to show at least one novel result.

Dear reviewer, we agree that we did not discover new functionality; the goal of this work was to engineer the control of a multicellular organism's physiology and gene expression with programmable bacteria. With mounting evidence of a microbiome's ability to influence its host, our

work is an innovative approach to bring programming control and logic (“AND/OR”) to these interactions.

For your comments, “I would expect the authors to engineer a new functionality into the organism and then control this functionality”, our response is that “engineer a new functionality” and “control this functionality” are two separate research directions, with the control of functionality being the focus of this work. The current projects going on in our lab include identifying the mechanisms behind bacteria-host interaction that changes host heat/cold tolerance or response to probiotics assumption. Although we have made some progress on these projects, these new functions/mechanisms would not fit into this paper, which is about new “control” but not about new “functionality”. Overall, we hope to have convinced you that new “control” and new “functions” are two different directions and that from synthetic biology perspective this new “control” over multicellular organisms is significant.

As stated in my initial review, I agree that the overall concept is interesting: if bacteria (and RNA expression) can be used as an intermediary to program a multicellular organism then it would be both novel and useful. However, the examples the authors show in the paper remain trivial and the authors do not in any meaningful way demonstrate that they can program behavior. Turning off a fluorescent protein, making an animal shake by knocking down a muscle protein, or increasing the fat content by reducing a transcription factor required for correct lipid metabolism do not constitute behaviors.

- Thank you for stating that the overall concept is interesting and that our system is novel and useful. We agree with your comment that we may have overreached on the term “behavior” since twitching could be a “behavior” or it could be more like a “physiology modulation”. With that, we have changed the title of our manuscript to “**Programming gene expression in multicellular organisms for physiology modulation through engineered bacteria**”. This title should cover our experiments in terms of turning off a fluorescent protein, switching on twitching behavior, and decreasing the fat content while not overreaching.

Furthermore, the experimental conditions (e.g., a poor choice of a gfp reporter with neuronal expression and an unrelated protein moiety attached) suggest to me that the presented results are best treated as pilot experiments that show promise. For example, the presented results would be appropriate for a grant proposal or a Ph.D. student pre-proposal exam.

- Thank you for the comments. We recognize that our choice of GFP reporter strain with neuronal expression was not the best platform to start with. Even though we didn’t start with the most ideal GFP target, we still achieved the right dynamic range with “ON/OFF” status. In reality, there is no guarantee that the multicellular organism gene that we will target is always the perfect target gene for RNAi assay. We expect that our “control” system can achieve multicellular organism modulation with both “perfect” and “imperfect” targets.

The text is easier to read but the authors still wildly overstate the implications of their observations. I do not see any coherent argument for the speculation that these advances can be used for agricultural or therapeutic purposes.

- Thank you for the comments. We have made the following changes to make sure we are not overstating the implications of our observations.

We have deleted lines 237-240: “this system can be used for disease diagnosis and therapeutics for mammals including human beings.”

We have also deleted the sentences in lines 241-146 “By fully exploring our platform, we could potentially reprogram insects and worm physiology and behaviors through bacterial RNA for RNAi toward agriculture applications. Meanwhile, by engineering bacteria with logic gates to recognize nutrients, disease biomarkers, gut positional signals, and orthogonal chemicals, we can potentially use our system for disease diagnosis and therapeutics delivery for mammals including human beings.”

We have also deleted the sentences in lines 58-61 “Using this platform as well as metabolite-based host-bacteria interactions, the programming of logic gates on the organism levels could enable exquisite control of biological events in insects, mammals, or humans, for any number of applications.”

In sum, I do not find that the methodologies or the results described can stand on their own as an independent manuscript. The methodology is not novel nor rigorously validated to an extent that it will open up for new experiments in other laboratories. My concerns about using RNAi as a general approach to program behavior due to **the slow response times** remain. The authors do not program behavior despite their claims. For the manuscript to be of general interest and useful for other laboratories, I would expect that the authors add a considerable amount of experimental results, some of which the authors describe as ongoing in their laboratory. I wish the authors had used the revision as an occasion to demonstrate the potential of the approach they describe.

- Thank you for the comments. As mentioned earlier, our novelty mainly lies in the new approach to “control” animal gene-expression/physiology through bacteria. This will inspire new thoughts over how to manipulate multicellular organisms, especially in the case when programming with logic “AND/OR” gates is beneficial. Because controlling and programming multicellular organisms is challenging, our approach presents a novel approach towards this goal.

- For your comment: “The authors do not program behavior despite their claims”, we have changed our title to “**Programming gene expression in multicellular organisms for physiology modulation through engineered bacteria**” to eliminate “behavior”. Also, as mentioned earlier, even if we produce more results in the lab for biological discoveries including how bacteria change animal physiology and their response to the environment. It is going to be different papers.

- About your comments that “RNAi as a general approach to program behavior is slow”. Utilizing microbe-animal relationships to modulate and program animals is relatively slower (response time in hours), but has the advantage of being much easier to manipulate and deliver (without genetic engineering of the animal directly) and is steadier. Being quick is not our goal for this work. Also, we do not think RNAi by feeding is the quickest approach for many other applications. For example, for mammals, metabolites/neuron transmitters are a better choice compared with RNAi. The potential approach to make the process faster is to use enzyme dimerization or protein degradation for metabolites production, which will be a much faster process than promoter induced transcription and translation. The goal of this paper is to demonstrate a proof of principle of our approach that will catalyze many future directions and applications.

Reviewer #3 (Remarks to the Author):

Most of the comments have been addressed sufficiently, except comments #1, 5 and 8. Dear reviewer, thank you for your comments. We appreciate your time and efforts. We have addressed your comments as follows.

Also, thank you for checking on Reviewer #2's comments and our responses. We appreciate Reviewer #2's comments on our manuscript, however, Reviewer #2 may come from a pure biology background and may not appreciate our innovation in synthetic biology. Reviewer #2 has been looking for new cellular functionality from our work while our manuscript focuses on new control modalities of multicellular organisms to program gene expression and physiology. Controlling and programming on multicellular organisms is challenging and our approach presents an exciting first step by enabling towards this goal. With that, we hope you will give another look at Reviewer #2's comments as well as our responses from a synthetic biology perspective.

Comment #1

(1) The introduction section is inadequate in highlighting the background and importance of RNA in host-microbe interaction. Please highlight the importance more clearly to justify the direction and scope of this study.

- We have added the following sentences in the introduction line 47 to highlight the importance of RNA in host-microbe interaction and to justify the direction and scope of this study. "RNA interference is a regulatory mechanism conserved in eukaryotes that play key roles in numerous biological processes, including RNA stability and processing, biotic and abiotic stress response, and the regulation of morphological and developmental events. Horizontal transfer of mobile RNAs between different species has been observed between pathogens/parasites and host animals, pathogens/parasites and host plants, and plants and animals. In *C. elegans*, RNAi is easy to implement because RNA can be delivered by feeding the worms bacteria that express double-stranded RNA complementary to a *C. elegans* gene of interest²⁸"

(2) While the elaboration is well-substantiated with relevant studies, the elaboration does not highlight RNAi as an intermediate in host-microbe interactions. Also, lines 237-240 seems to be an overreach: "this system can be used for disease diagnosis and therapeutics for mammals including human beings." Physiologically, *C. elegans* is not the most relevant system for disease diagnosis and therapeutics delivery in mammals. It would be better to provide a more nuanced elaboration for human application.

- Thank you for your comments.

We agree that line 237-240 is an overreach and thus has deleted the whole sentence. In this paper, we should focus on multi-cellular organism gene-expression regulation and thus has changed the title to "**Programming gene expression in multicellular organisms for physiology modulation through engineered bacteria**" to make sure we don't overreach in terms of potential applications.

We have also deleted the sentences in lines 241-146 "By fully exploring our platform, we could potentially reprogram insects and worm physiology and behaviors through bacterial RNA for RNAi toward agriculture applications. Meanwhile, by engineering bacteria with logic gates to recognize nutrients, disease biomarkers, gut positional signals, and orthogonal chemicals, we can potentially use our system for disease diagnosis and therapeutics delivery for mammals including human beings."

We have also deleted the sentences in lines 58-61 "Using this platform as well as metabolite-based host-bacteria interactions, the programming of logic gates on the organism levels could enable exquisite control of biological events in insects, mammals, or humans, for any number of applications."

Comment #5

(1) The optimization data used to establish the combination of inducer concentrations to demonstrate OR and AND gates are not provided. In the same context, the arabinose concentrations used for AND gate and OR gate are significantly different. Please provide data to justify the inducer concentrations used in this assay.

(2) The caption for the newly revised Figure S5 does not adequately describe the figure.
- We have added the *unc-22* and *sbp-1* RNA length and inducer concentration information in the manuscript and put this information in the Fig. S5 and S6 caption.

Figure S5. Optimization of RNA lengths and inducer concentrations. **(A)** Optimization of *unc-22* RNA length and format for inducing *C. elegans* twitching behavior by **1 mM IPTG**. The fraction of twitched worms were counted for worms fed on BL21(DE3) synthesizing double or single-stranded *unc-22* RNA with lengths 100, 200, and 400 bp. Both 200 bp and 400 bp *unc-22* dsRNA and ssRNA achieved “on” and “off” modulation of worm twitching behavior **(B)** Length optimization of *unc-22* RNA for *C. elegans* twitching behavior controlled by AND gate. BL21 co-transformed with the AND-gate circuit plasmid and the indicated RNA synthesizing plasmid were fed to N2 worms with **no inducer (control)**, **0.2 mg/mL L-arabinose (Ara)**, **0.1 µg/mL anhydrotetracycline (aTc)**, or **both inducers**. Bacteria producing ds400 led to complete twitching behavior in *C. elegans* with both inducers, but also caused twitching behavior in around 70% of the worms with **0.2 mg/mL Ara** only sample. Therefore, the Ara concentration was further optimized with ds400 RNA and **0.1 µg/mL aTc** in *C.* **(C)** Inducer concentration optimization for *C. elegans* twitching behavior controlled by AND gate. A series of Ara concentration was used (indicated on the x-axis) in combination with **0.1 µg/mL aTc**. As a result, **2 µg/mL Ara** and **0.1 µg/mL aTc** together were able to achieve full “ON/OFF” status with AND-gate and ds400 *unc-22* RNA for *C. elegans* twitching behavior control. **(D)** Length optimization for OR gate on *C. elegans* twitching. BL21 co-transformed with the OR-gate circuit plasmid and RNA synthesizing plasmid were fed to N2 worms with **no inducer (control)**, **0.2 mg/mL Ara**, **0.1 µg/mL aTc**, or **both inducers**. Since our initial try with both ds400 and ss400 showed the OR-gate profile using **0.2 mg/mL Ara** and **0.1 µg/mL aTc**, the inducer concentrations were not further optimized. (Error bars show the mean ± standard deviation for n=15).

Figure S6. Optical and fluorescence microscopy images of *C. elegans* fat storage. **(A)** OR-gate controlled *C. elegans* fat storage indicated by Nile Red staining in response to **0.2 mg/mL of L-arabinose (Ara)**, **0.1 µg/mL of anhydrotetracycline (aTc)**, or both inducers. Since our initial try with full-length *sbp-1*, **0.2 mg/mL Ara** and **0.1 µg/mL aTc** worked, the inducer concentrations and RNA length were not further optimized. **(B)** AND-gate controlled *C. elegans* fat storage in response **2 µg/mL Ara**, **0.1 µg/mL of aTc**, or both inducers. **(C)** Inducer concentration optimization for *C. elegans* fat-storage programmed by AND gate. The initial AND-gate controlled fat-storage with **0.2 mg/mL Ara** and **0.1 µg/mL aTc** didn't work because 0.2 mg/mL of Ara alone led to a similar level of silencing effect as with both inducers. To optimize the inducer concentration, Ara concentration was diluted to **20 µg/mL** and **2 µg/mL** while aTc concentration was kept constant at **0.1 µg/mL**. In the end, **2 µg/mL Ara** and **0.1 µg/mL aTc** were picked for the AND-gate experiments. Since full-length *sbp-1* worked after inducer concentration optimization, RNA length experiments were not further conducted. (Error bars show the mean ± standard deviation for n=5)

Comment #8

(1) Screening and validation for *sbp-1* RNAi candidates are lacking, similar to *unc-22*. Please provide the required data.

(2) The reasoning for not conducting RNA length experiments is confusing and not convincing. From the revised manuscript, *unc-22* RNA length screening was done as shown in Figure S5. Are authors suggesting that RNA length experiments for *sbp-1* were not conducted because the dynamic range/difference between 'on' and 'off' states for *sbp-1* was distinguishable? If so, please explain this clearly.

- Thank you for your comments on the RNA length experiments. You are 100% correct about the *sbp-1* RNA length experiment. Since our initial try with full-length *sbp-1* gave a good dynamic range between "on" and "off" states, we did not conduct the RNA length experiments for *sbp-1*. We have thus added this sentence in line 177-179. "Because full length *sbp-1* RNA modulated *C. elegans* fat storage with a clear on and off status after inducer concentration optimization (Fig. S6C), RNA length optimization experiments were not performed as in the *unc-22* experiment."

REVIEWER COMMENTS

Reviewer #3 (Remarks to the Author):

Most of the comments have been addressed sufficiently.

Comment #1 - sufficiently addressed.

Comment #5 - sufficiently addressed. A minor suggestion to authors is to provide inducer concentration for Figure S5C and S6C.

Comment #8 - sufficiently addressed. A minor suggestion to authors is to capitalize 'ON' and 'OFF' states in lines 177-179 to standardize the formatting.

Reviewer #3 (Remarks to the Author):

Most of the comments have been addressed sufficiently.

Comment #1 - sufficiently addressed.

Comment #5 - sufficiently addressed. A minor suggestion to authors is to provide inducer concentration for Figure S5C and S6C.

Dear reviewer, thank you for the suggestion. We have updated Figure S5C and S6C to include the inducer concentrations.

Comment #8 - sufficiently addressed. A minor suggestion to authors is to capitalize 'ON' and 'OFF' states in lines 177-179 to standardize the formatting.

We have capitalized all 'ON' and 'OFF' throughout the manuscript to standardize the formatting.

Line 75-77: "We started with BL21 because the relatively lower dsRNA production is correlated with less RNAi efficiency in *C. elegans* and thus can potentially provide a dynamic range of RNAi for a complete "ON" and OFF" state."

Line 95-97: "After feeding the bacteria containing these plasmids to the worms, we found that single-stranded RNA sequences of 200bp, 300bp, and 400 bp achieved distinguishable "ON" and "OFF" status of GFP expression in *C. elegans* with and without IPTG (Fig. 2B)."

Line 109-111: "Since neuronal genes are more difficult to target with RNAi³⁶, we usually see around 20% residual fluorescence on our "OFF" stage, which should come from the neuronal GFP that were not RNAi silenced."

Line 175-177: "Because full length sbp-1 RNA modulated *C. elegans* fat storage with a clear "ON" and "OFF" status after inducer concentration optimization (Fig. S6C), RNA length optimization experiments were not performed as in the *unc-22* experiment."